# Towards Safe Policy Improvement for Non-Stationary MDPs

**Yash Chandak**
University of Massachusetts
ychandak@cs.umass.edu

**Scott M. Jordan**
University of Massachusetts
sjordan@cs.umass.edu

**Georgios Theocharous**
Adobe Research
theochar@adobe.com

**Martha White**
University of Alberta & Amii
whitem@alberta.ca

**Philip S. Thomas**
University of Massachusetts
pthomas@cs.umass.edu

## Abstract

Many real-world sequential decision-making problems involve critical systems with financial risks and human-life risks. While several works in the past have proposed methods that are *safe* for deployment, they assume that the underlying problem is *stationary*. However, many real-world problems of interest exhibit non-stationarity, and when stakes are high, the cost associated with a false stationarity assumption may be unacceptable. We take the first steps towards ensuring safety, with high confidence, for smoothly-varying non-stationary decision problems. Our proposed method extends a type of safe algorithm, called a *Seldonian algorithm*, through a synthesis of model-free reinforcement learning with time-series analysis. Safety is ensured using sequential hypothesis testing of a policy's *forecasted* performance, and confidence intervals are obtained using *wild bootstrap*.

## 1   Introduction

Reinforcement learning (RL) methods have been applied to real-world sequential decision-making problems such as diabetes management [5], sepsis treatment [50], and budget constrained bidding [66]. For such real-world applications, safety guarantees are critical to mitigate serious risks in terms of both human-life and monetary assets. More concretely, here, by *safety* we mean that any update to a system should not reduce the performance of an existing system (e.g., a doctor's initially prescribed treatment). A further complication is that these applications are non-stationary, violating the foundational assumption [53] of *stationarity* in most RL algorithms. This raises the main question we aim to address: *How can we build sequential decision-making systems that provide safety guarantees for problems with non-stationarities?*

Conventionally, RL algorithms designed to ensure safety [47, 26, 58, 69, 38, 17] model the environment as a Markov decision process (MDP), and rely upon the *stationarity assumption* made by MDPs [53]. That is, MDPs assume that a decision made by an *agent* always results in the same (distribution of) consequence(s) when the environment is in a given state. Consequently, safety is only ensured by prior methods when this assumption holds, which is rare in real-world problems.

While some works for lifelong-reinforcement learning [10, 1, 12, 13] or meta-reinforcement learning [3, 67] do aim to address the problem of non-stationarity, they do not provide any safety guarantees. Perhaps the work most closely related to ours is by Ammar et al. [4], which aims to find a policy that satisfies a safety constraint in the lifelong-learning setting. They use a follow-the-regularized-leader (FTRL) [51] approach to first perform an unconstrained maximization over the *average* performance over all the trajectories collected in the past, and then project the resulting solution onto a safe

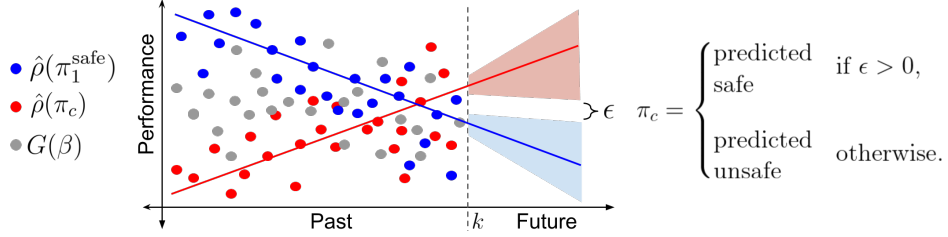

Figure 1: An illustration of the proposed idea where *safety* is defined to ensure that the future performance of a proposed policy $\pi_c$ is never worse than that of an existing, known, safe policy $\pi^{\text{safe}}$. The gray dots correspond to the returns, $G(\beta)$, observed for a policy $\beta$. The red and the blue dots correspond to the counterfactual estimates, $\hat{\rho}(\pi_c)$ and $\hat{\rho}(\pi^{\text{safe}})$, for performance of $\pi_c$ and $\pi^{\text{safe}}$, respectively. The shaded regions correspond to the uncertainty in future performance obtained by analysing the trend of the counterfactual estimates for past performances.

set. However, as shown by Chandak et al. [13], FTRL based methods can suffer from a significant performance lag in non-stationary environments. Further, the parameter projection requires *a priori* knowledge of the set of safe policy parameters, which might be infeasible to obtain for many problems, especially when the constraint is to improve performance over an existing policy or when the safe set is non-convex (e.g., when using policies parameterized using neural networks). Additionally, the method proposed by Chandak et al. [13] for policy improvement does not provide safety guarantees, and thus it would be irresponsible to apply it to safety-critical problems.

**Contributions:** In this work, we formalize the *safe policy improvement* problem for a more realistic non-stationary setting and provide an algorithm for addressing it. Additionally, a user-controllable knob is provided to set the desired *confidence level*: the maximum admissible probability that a worse policy will be deployed. The proposed method relies only on estimates of future performance, with associated confidence intervals. It does not require building a model of a non-stationary MDP (NS-MDP), and so it is applicable to a broader class of problems, as modeling an NS-MDP can often be prohibitively difficult. In Figure 1, we provide an illustration of the proposed approach for ensuring safe policy improvement for NS-MDPs.

**Limitations:** The method that we propose is limited to settings where both (a) non-stationarity is governed by an exogenous process (that is, past actions do not impact the underlying non-stationarity), and (b) the performance of every policy changes smoothly over time and has no discontinuities (abrupt breaks or jumps). Further, the use importance sampling makes our method prone to high variance.

## 2 Notation

We represent an NS-MDP as a *stochastic sequence*, $\{M_i\}_{i=1}^\infty$, of stationary MDPs $M_i \in \mathcal{M}$, where $\mathcal{M}$ is the set of all stationary MDPs. Each $M_i$ is a tuple $(\mathcal{S}, \mathcal{A}, \mathcal{P}_i, \mathcal{R}_i, \gamma, d^0)$, where $\mathcal{S}$ is the set of possible states, $\mathcal{A}$ is the set of actions, $\gamma \in [0,1)$ is the *discounting factor* [53], $d^0$ is the start-state distribution, $\mathcal{R}_i : \mathcal{S} \times \mathcal{A} \to \Delta(\mathbb{R})$ is the reward distribution, and $\mathcal{P}_i : \mathcal{S} \times \mathcal{A} \to \Delta(\mathcal{S})$ is the transition function, where $\Delta$ denotes a conditional distribution. For all $M_i \in \mathcal{M}$, we assume that $\mathcal{S}, \mathcal{A}, \gamma$, and $d^0$ remain fixed. We represent a policy as $\pi : \mathcal{S} \to \Delta(\mathcal{A})$.

Let $s_\infty$ be a *terminal absorbing state* [53] and an *episode* be a sequence of interactions with a given MDP, which enters $s_\infty$ within $T$ time steps. In general, we will use subscripts to denote the episode number and superscripts to denote the time step within an episode. That is, $S_i^t, A_i^t$, and $R_i^t$ are the random variables corresponding to the state, action, and reward at time step $t$ in episode $i$. Let a trajectory for episode $i$ generated using a policy (also known as a *behavior policy*) $\beta_i$ be $H_i := \{S_i^j, A_i^j, \beta_i(A_i^j|S_i^j), R_i^j\}_{j=0}^\infty$, where $\forall(i,t),\ R_i^t \in [-R_{\max}, R_{\max}]$. Let a *return* of $\pi$ for any $m \in \mathcal{M}$ be $G(\pi, m) := \sum_{t=0}^\infty \gamma^t R^t$ and the *expected return* $\rho(\pi, m) := \mathbb{E}[G(\pi, m)]$. With a slight overload of notation, let the *performance* of $\pi$ for episode $i$ be $\rho(\pi, i) := \mathbb{E}[\rho(\pi, M_i)]$. We will use $k$ to denote the most recently finished episode, such that episode numbers $[1, k]$ are in the past and episode numbers $(k, \infty]$ are in the future.

# 3 The Safe Policy Improvement Problem for Non-stationary MDPs

In this section, we formally present the problem statement, discuss the difficulty of this problem, and introduce a smoothness assumption that we leverage to make the problem tractable.

**Problem Statement:** Let $\mathcal{D} := \{(i, H_i) : i \in [1, k]\}$ be a random variable denoting a set of trajectories observed in the past and let alg be an algorithm that takes $\mathcal{D}$ as input and returns a policy $\pi$. Let $\pi^{\text{safe}}$ be a known safe policy, and let $(1 - \alpha) \in [0, 1]$ be a constant selected by a user of alg, which we call the *safety level*. We aim to create an algorithm alg that ensures with high probability that alg$(\mathcal{D})$, the policy proposed by alg, does not perform worse than the existing safe policy $\pi^{\text{safe}}$ during the *future* episode $k + \delta$. That is, we aim to ensure the following *safety guarantee*,

$$\Pr\left(\rho(\text{alg}(\mathcal{D}), k + \delta) \geq \rho(\pi^{\text{safe}}, k + \delta)\right) \geq 1 - \alpha. \tag{1}$$

**Hardness of the Problem:** While it is desirable to ensure the safety guarantee in (1), obtaining a new policy from alg$(\mathcal{D})$ that meets the requirement in (1) might be impossible unless some more regularity assumptions are imposed on the problem. To see why, notice that if the environment can change arbitrarily, then there is not much hope of estimating $\rho(\pi, k + \delta)$ accurately since $\rho(\pi, k + \delta)$ for any $\pi$ could be any value between the extremes of all possible outcomes, regardless of the data collected during episodes 1 through $k$.

To avoid arbitrary changes, previous works typically require the transition function $\mathcal{P}_k$ and the reward function $\mathcal{R}_k$ to be Lipschitz smooth over time [39, 30, 40, 16]. And in fact, we can provide a bound on the change in performance given such Lipschitz conditions, as we show below in Theorem 1. Unfortunately, this bound is quite large: unless the Lipschitz constants are so small that they effectively make the problem stationary, the performance of a policy $\pi$ across consecutive episodes can still fluctuate wildly. Notice that due to the inverse dependency on $(1 - \gamma)^2$, if $\gamma$ is close to one, then the Lipschitz constant $L$ can be enormous even when $\epsilon_P$ and $\epsilon_R$ are small. In Appendix B, we provide an example of an NS-MDP for which Theorem 1 holds with exact equality, illustrating that the bound is tight.

**Theorem 1** (Lipschitz smooth performance). *If $\exists \epsilon_P \in \mathbb{R}$ and $\exists \epsilon_R \in \mathbb{R}$ such that for any $M_k$ and $M_{k+1}$, $\forall s \in \mathcal{S}, \forall a \in \mathcal{A}$, $\|\mathcal{P}_k(\cdot|s, a) - \mathcal{P}_{k+1}(\cdot|s, a)\|_1 \leq \epsilon_P$ and $|\mathbb{E}[\mathcal{R}_k(s, a)] - \mathbb{E}[\mathcal{R}_{k+1}(s, a)]| \leq \epsilon_R$, then the performance of any policy $\pi$ is Lipschitz smooth over time, with Lipschitz constant $L := \left(\frac{\gamma R_{max}}{(1-\gamma)^2}\epsilon_P + \frac{1}{1-\gamma}\epsilon_R\right)$. That is, $\forall k \in \mathbb{N}_{>0}, \forall \delta \in \mathbb{N}_{>0}$, $|\rho(\pi, k) - \rho(\pi, k + \delta)| \leq L\delta$.*

All proofs are deferred to the appendix.

**An alternate assumption:** In many real-world sequential decision-making problems, while there is non-stationarity, the performance of a policy $\pi$ does not fluctuate wildly between consecutive episodes. Examples where performance changes are likely more regular include the effect of a medical treatment on a patient; the usefulness of online recommendations based on the interests of a user; or the quality of a controller as a robot's motor friction or battery capacity degrades. Therefore, instead of considering smoothness constraints on the transition function $\mathcal{P}_k$ and the reward function $\mathcal{R}_k$ like above, we consider more direct smoothness constraints on the performance $\rho(\pi, i)$ of a policy $\pi$. Similar assumptions have been considered for analyzing trends for digital marketing [60] and remain popular among policymakers for designing policies based on forecasting [64].

If $\rho(\pi, i)$ changes smoothly with episode $i$, then the performance trend of a given policy $\pi$ can be seen as a *univariate time-series*, i.e., a sequence of *scalar* values corresponding to performances $\{\rho(\pi, i)\}_{i=1}^{k}$ of $\pi$ during episodes 1 to $k$. Leveraging this observation, we propose modeling the performance trend using a linear regression model that takes an episode number as the input and provides a performance prediction as the output. To ensure that a wide variety of trends can be modeled, we use a $d$-dimensional non-linear *basis function* $\phi : \mathbb{N}_{>0} \to \mathbb{R}^{1 \times d}$. For example, $\phi$ can be the Fourier basis, which has been known to be useful for modeling a wide variety of trends and is fundamental for time-series analysis [9]. We state this formally in the following assumption,

**Assumption 1** (Smooth performance). *For every policy $\pi$, there exists a sequence of mean-zero and independent noises $\{\xi_i\}_{i=1}^{k+\delta}$, and $\exists w \in \mathbb{R}^{d \times 1}$, such that, $\forall i \in [1, k + \delta]$, $\rho(\pi, M_i) = \phi(i)w + \xi_i$.*

Recall that the stochasticity in $\rho(\pi, M_i)$ is a manifestation of stochasticity in $M_i$, and thus this assumption requires that the performance of $\pi$ during episode $i$ is $\rho(\pi, i) = \mathbb{E}[\rho(\pi, M_i)] = \phi(i)w$.

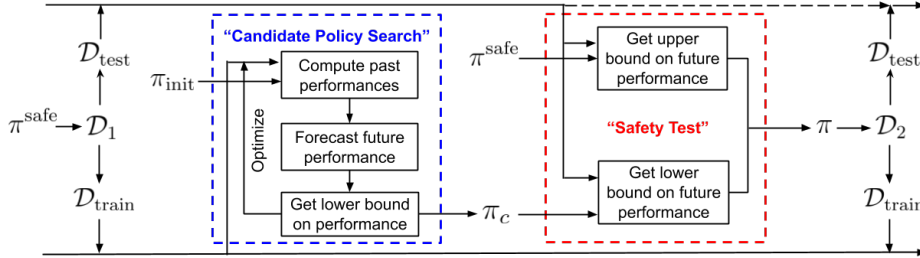

Figure 2: The proposed algorithm first partitions the initial data $\mathcal{D}_1$ into two sets, namely $\mathcal{D}_{\text{train}}$ and $\mathcal{D}_{\text{test}}$. Subsequently, $\mathcal{D}_{\text{train}}$ is used to search for a possible *candidate policy* $\pi_c$ that might improve the future performance, and $\mathcal{D}_{\text{test}}$ is used to perform a safety test on the proposed candidate policy $\pi_c$. The existing safe policy $\pi^{\text{safe}}$ is only updated if the proposed policy $\pi_c$ passes the safety test.

Assumption 1 is reasonable for several reasons. The first is that the noise assumptions are not restrictive. The distribution of $\xi_i$ does not need to be known and the $\xi_i$ can be non-identically distributed. Additionally, both $w$ and $\{\xi_i\}_{i=1}^{k+\delta}$ can be different for different policies. The independence assumption only states that at each time step, the variability in performance due to sampling $M_i$ is independent of the past (i.e., there is no auto-correlated noise).

The strongest requirement is that the performance trend be a linear function of the basis $\phi$; but because $\phi$ is a generic basis, this is satisfied for a large set of problems. Standard methods that make stationarity assumptions correspond to our method with $\phi(s) = [1]$ (fitting a horizontal line). Otherwise, $\phi$ is generic: we might expect that there exist sufficiently rich features (e.g., Fourier basis [9])) for which Assumption 1 is satisfied. In practice, we may not have access to such a basis, but like any time-series forecasting problem, goodness-of-fit tests [15] can be used by practitioners to check whether Assumption 1 is reasonable before applying our method.

The basis requirement, however, can be a strong condition and could be violated. This assumption is *not* applicable for settings where there are jumps or breaks in the performance trend. For example, performance change is sudden when a robot undergoes physical damage, its sensors are upgraded, or it is presented with a completely new task. The other potential violation is the fact that the basis is a function of time. Since the dimension $d$ of the basis $\phi$ is finite and fixed, but $k$ can increase indefinitely, this assumption implies that performance trends of the policies must exhibit a global structure, such as periodicity. This can be relaxed using auto-regressive methods that are better at adapting to the local structure of any time-series. We discuss this and other potential future research directions in Section 10.

## 4 SPIN: Safe Policy Improvement for Non-Stationary Settings

To ensure safe policy improvement, we adapt the generic template of the Seldonian framework [61] to the non-stationary setting. The overall approach consists of continually (1) taking an existing safe policy; (2) finding a candidate policy that has (reasonably high) potential to be a strict improvement on the safe policy; (3) testing if this candidate policy is still safe and is an improvement with high confidence; (4) updating the policy to be the candidate policy only if it passes the test; and (5) gathering more data with the current safe policy to get data for the next candidate policy search. This procedure consists of four key technical steps: *performance estimation*, *safety test*, *candidate policy search*, and *data-splitting*. A schematic diagram of the overall procedure is provided in Figure 2.

**Performance Estimation:** To develop an algorithm that ensures the safety constraint in (1), we first require an estimate $\hat{\rho}(\pi, k + \delta)$ of the future performance $\rho(\pi, k + \delta)$ and the uncertainty of this estimate, namely a function $\mathscr{C}$ for obtaining a confidence interval (CI) on future performances. Under Assumption 1, estimating $\rho(\pi, k + \delta)$ (the performance of a policy $\delta$ episodes into the future) can be seen as a *time-series forecasting* problem given the *performance trend* $\{\rho(\pi, i)\}_{i=1}^{k}$. We build upon the work by Chandak et al. [13] to estimate $\rho(\pi, k + \delta)$. However, to the best of our knowledge, no method yet exists to obtain $\mathscr{C}$. A primary contribution of this work is to provide a solution to this technical problem, developed in Section 5.

**Safety Test:** To satisfy the required safety constraint in (1), an algorithm `alg` needs to ensure with high-confidence that a given $\pi_c$, which is a *candidate policy* for updating the existing safe policy $\pi^{\text{safe}}$, will have a higher future performance than that of $\pi^{\text{safe}}$. Importantly, just as the future performance, $\rho(\pi_c, k + \delta)$, of $\pi_c$ is not known *a priori* for a non-stationary MDP, the future performance of the baseline policy $\pi^{\text{safe}}$ is also not known *a priori*. Therefore, to ensure that the constraint in (1) is satisfied, we use $\mathscr{C}$ to obtain a *one-sided* lower and upper bound for $\rho(\pi_c, k + \delta)$ and $\rho(\pi^{\text{safe}}, k + \delta)$, respectively, each with confidence level $\alpha/2$. The confidence level is set to $\alpha/2$ so that the total failure rate (i.e., either $\rho(\pi_c, k + \delta)$ or $\rho(\pi^{\text{safe}}, k + \delta)$ is outside their respective bounds) is no more than $\alpha$. Subsequently, `alg` only updates $\pi^{\text{safe}}$ if the lower bound of $\rho(\pi_c, k + \delta)$ is higher than the upper bound of $\rho(\pi^{\text{safe}}, k + \delta)$; otherwise, no update is made and $\pi^{\text{safe}}$ is chosen to be executed again.

**Candidate Policy Search:** An ideal candidate policy $\pi_c$ would be one that has high future performance $\rho(\pi_c, k + \delta)$, along with a large confidence lower bound on its performance, so that it can pass the safety test. However, in practice, there could often be conflicts between policies that might have higher estimated future performance but with lower confidence, and policies with lower estimates of future performance but with higher confidence. As the primary objective of our method is to ensure safety, we draw inspiration from prior methods for conservative/safe learning in stationary domains [26, 57, 34, 17] and propose searching for a policy that has the *highest lower* confidence bound. That is, let the one-sided CI for the future performance $\rho(\pi, k + \delta)$ obtained using $\mathscr{C}$ be $[\hat{\rho}^{\text{lb}}(\pi), \infty)$, then $\pi_c \in \arg\max_\pi \hat{\rho}^{\text{lb}}(\pi)$.

**Data-Splitting:** Conventionally, in the time-series literature, there is only a single trend that needs to be analyzed. In our problem setup, however, the time series forecasting function is used to analyze trends of multiple policies during the candidate policy search. If all of the available data $\mathcal{D}$ is used to estimate the lower bound $\hat{\rho}^{\text{lb}}(\pi)$ for $\rho(\pi, k + \delta)$ and if $\pi$ is chosen by maximizing $\hat{\rho}^{\text{lb}}(\pi)$, then due to the *multiple comparisons problem* [6] we are likely to find a $\pi$ that over-fits to the data and achieves a higher value of $\hat{\rho}^{\text{lb}}(\pi)$. A safety test based on such a $\hat{\rho}^{\text{lb}}(\pi)$ would thus be unreliable. To address this problem, we partition $\mathcal{D}$ into two mutually exclusive sets, namely $\mathcal{D}_{\text{train}}$ and $\mathcal{D}_{\text{test}}$, such that only $\mathcal{D}_{\text{train}}$ is used to search for a candidate policy $\pi_c$ and only $\mathcal{D}_{\text{test}}$ is used during the safety test.

## 5  Estimating Confidence Intervals for Future Performance

To complete the SPIN framework discussed in Section 4, we need to obtain an estimate $\hat{\rho}(\pi, k + \delta)$ of $\rho(\pi, k + \delta)$ and its confidence interval using the function $\mathscr{C}$. This requires answering two questions: (1) Given that in the past, policies $\{\beta_i\}_{i=1}^k$ were used to generate the observed returns, how do we estimate $\hat{\rho}(\pi, k + \delta)$ for a *different* policy $\pi$? (2) Given that the trajectories are obtained only from a *single* sample of the sequence $\{M_i\}_{i=1}^k$, how do we obtain a confidence interval around $\hat{\rho}(\pi, k + \delta)$? We answer these two questions in this section.

**Point Estimate of Future Performance:** To answer the first question, we build upon the following observation used by Chandak et al. [13]: While in the past, returns were observed by executing policies $\{\beta_i\}_{i=1}^k$, *what if* policy $\pi$ was executed instead?

Formally, we use per-decision importance sampling [48] for $H_i$, to obtain a *counterfactual* estimate $\hat{\rho}(\pi, i) \coloneqq \sum_{t=0}^{\infty} \left( \prod_{l=0}^{t} \frac{\pi(A_i^l | S_i^l)}{\beta_i(A_i^l | S_i^l)} \right) \gamma^t R_i^t$, of $\pi$'s performance in the past episodes $i \in [1, k]$. This estimate $\hat{\rho}(\pi, i)$ is an unbiased estimator of $\rho(\pi, i)$, i.e., $\mathbb{E}[\hat{\rho}(\pi, i)] = \rho(\pi, i)$, under the the following assumption [58], which can typically be satisfied using an entropy-regularized policy $\beta_i$.

**Assumption 2** (Full Support). $\forall a \in \mathcal{A}$ and $\forall s \in \mathcal{S}$ there exists a $c > 0$ such that $\forall i, \beta_i(a|s) > c$.

Having obtained counterfactual estimates $\{\hat{\rho}(\pi, i)\}_{i=1}^k$, we can then estimate $\rho(\pi, k + \delta)$ by analysing the performance trend of $\{\hat{\rho}(\pi, i)\}_{i=1}^k$ and forecasting the future performance $\hat{\rho}(\pi, k + \delta)$. That is, let $X \coloneqq [1, 2, ..., k]^\top \in \mathbb{R}^{k \times 1}$, let $\Phi \in \mathbb{R}^{k \times d}$ be the corresponding basis matrix for $X$ such that $i^{\text{th}}$ row of $\Phi$, $\forall i \in [1, k]$, is $\Phi_i \coloneqq \phi(X_i)$, and let $Y \coloneqq [\hat{\rho}(\pi, 1), \hat{\rho}(\pi, 2), ..., \hat{\rho}(\pi, k)]^\top \in \mathbb{R}^{k \times 1}$. Then under Assumptions 1 and 2, an estimate $\hat{\rho}(\pi, k + \delta)$ of the future performance can be computed using least-squares (LS) regression, i.e., $\hat{\rho}(\pi, k + \delta) = \phi(k + \delta)\hat{w} = \phi(k + \delta)(\Phi^\top \Phi)^{-1} \Phi^\top Y$.

**Confidence Intervals for Future Performance:**     We now aim to quantify the uncertainty of $\hat{\rho}(\pi, k + \delta)$ using a confidence interval (CI), such that the true future performance $\rho(\pi, k + \delta)$ will be contained within the CI with the desired confidence level. To obtain a CI for $\rho(\pi, k + \delta)$, we

make use of t-statistics [62] and use the following notation. Let the sample standard deviation for $\hat{\rho}(\pi, k + \delta)$ be $\hat{s}$, where $\hat{s}^2 := \phi(k+\delta)(\Phi^\top\Phi)^{-1}\Phi^\top\hat{\Omega}\Phi(\Phi^\top\Phi)^{-1}\phi(k+\delta)^\top$, where $\hat{\Omega}$ is a diagonal matrix containing the square of the regression errors $\hat{\xi}$ (see Appendix C.1 for more details), and let the t-statistic be $\text{t} := (\hat{\rho}(\pi, k+\delta) - \rho(\pi, k+\delta))/\hat{s}$.

If the distribution of t was known, then a $(1 - \alpha)100\%$ CI could be obtained as $[\hat{\rho}(\pi, k + \delta) - \hat{s}\text{t}_{1-\alpha/2}, \ \hat{\rho}(\pi, k + \delta) - \hat{s}\text{t}_{\alpha/2}]$, where for any $\alpha \in [0, 1], \text{t}_\alpha$ represents $\alpha$-*percentile* of the t distribution. Unfortunately, the distribution of t is not known. One alternative could be to assume that t follows the *student-*t distribution [52]. However, that would only be valid if *all* the error terms in regression are *homoscedastic* and *normally* distributed. Such an assumption could be severely violated in our setting due to the heteroscedastic nature of the estimates of the past performances resulting from the use of potentially different behavior policies $\{\beta_i\}_{i=1}^k$ and due to the unknown form of stochasticity in $\{M_i\}_{i=1}^k$. Further, due to the use of importance sampling, the performance estimates $\{\hat{\rho}(\pi, i)\}_{i=1}^k$ can often be skewed and have a heavy-tailed distribution with high-variance [59]. We provide more discussion on these issues in Appendix D.3.

To resolve the above challenges, we make use of *wild bootstrap*, a semi-parametric bootstrap procedure that is popular in time series analysis and econometrics [65, 41, 43, 19, 20]. The idea is to generate multiple pseudo-samples of performance for each $M_i$, using the single sampled performance estimate. These multiple pseudo-samples can then be used to obtain an empirical distribution and so characterize the range of possible performances. This empirical distribution is for a pseudo t-statistic, $\text{t}^*$, where the $\alpha$-percentile for $\text{t}^*$ can be used to estimate the $\alpha$-percentile of the distribution of t. Below, we discuss how to get these multiple pseudo-samples.

Recall that trajectories $\{H_i\}_{i=1}^k$ are obtained only from a *single* sample of the sequence $\{M_i\}_{i=1}^k$. Due to this, only a single point estimate $\hat{\rho}(\pi, k + \delta)$, devoid of any estimate of uncertainty, of the future performance $\rho(\pi, k + \delta)$ can be obtained. Therefore, we aim to create *pseudo-samples* of $\{\hat{\rho}(\pi, i)\}_{i=1}^k$ that *resemble* the estimates of past performances that would have been obtained using trajectories from an alternate sample of the sequence $\{M_i\}_{i=1}^k$. The wild bootstrap procedure provides just such an approach, with the following steps.

1. Let $Y^+ := [\rho(\pi, 1), ..., \ \rho(\pi, k)]^\top \in \mathbb{R}^{k \times 1}$ correspond to the true performances of $\pi$. Create $\hat{Y} = \Phi(\Phi^\top\Phi)^{-1}\Phi^\top Y$, an LS estimate of $Y^+$, using the counterfactual performance estimates $Y$ and obtain the regression errors $\hat{\xi} := \hat{Y} - Y$.
2. Create pseudo-noises $\xi^* := \hat{\xi} \odot \sigma^*$, where $\odot$ represents Hadamard product and $\sigma^* \in \mathbb{R}^{k \times 1}$ is Rademacher random variable (i.e., $\forall i \in [1, k], \ \Pr(\sigma_i^* = +1) = \Pr(\sigma_i^* = -1) = 0.5$).[1]
3. Create pseudo-performances $Y^* := \hat{Y} + \xi^*$, to obtain pseudo-samples for $\hat{Y}$ and $\hat{\rho}(\pi, k + \delta)$ as $\hat{Y}^* = \Phi(\Phi^\top\Phi)^{-1}\Phi^\top Y^*$ and $\hat{\rho}(\pi, k + \delta)^* = \phi(k + \delta)(\Phi^\top\Phi)^{-1}\Phi^\top Y^*$.

Steps 2 and 3 can be repeated to re-sample up to $B \leq 2^k$ *similar* sequences of past performance $Y^*$, from a *single* observed sequence $Y$ of length $k$, while also preserving the time-series structure. This unreasonable property led Mammen [43] to coin the term 'wild bootstrap'. For a brief discussion on *why* wild bootstrap works, see Appendix D.1.

Given these multiple pseudo-samples, we can now obtain an empirical distribution for pseudo t-statistic, $\text{t}^*$. Let the pseudo-sample standard deviation be $\hat{s}^*$, where $\hat{s}^{*2} := \phi(k + \delta)(\Phi^\top\Phi)^{-1}\Phi^\top\hat{\Omega}^*\Phi(\Phi^\top\Phi)^{-1}\phi(k+\delta)^\top$, where $\hat{\Omega}^*$ is a diagonal matrix containing the square of the pseudo-errors $\hat{\xi}^* := \hat{Y}^* - Y^*$. Let $\text{t}^* := (\hat{\rho}(\pi, k + \delta)^* - \hat{\rho}(\pi, k + \delta))/\hat{s}^*$. Then an $\alpha$-percentile $\text{t}_\alpha^*$ of the empirical distribution of $\text{t}^*$ is used to estimate the $\alpha$-percentile of t's distribution.

Finally, we can define $\mathscr{C}$ to use the wild bootstrap to produce CIs. To ensure this is principled, we leverage a property proven by Djogbenou et al. [23] and show in the following theorem that the CI for $\rho(\pi, k + \delta)$ obtained using pseudo-samples from wild bootstrap is *consistent*. For simplicity, we restrict our focus to settings where $\phi$ is the Fourier basis (see Appendix C.2 for more discussion).

**Theorem 2** (Consistent Coverage). *Under Assumptions 1 and 2, if the trajectories $\{H_i\}_{i=1}^k$ are independent and if $\phi(x)$ is a Fourier basis, then as $k \to \infty$,*

$$\Pr\left(\rho(\pi, k + \delta) \in \left[\hat{\rho}(\pi, k + \delta) - \hat{s}\,t_{1-\alpha/2}^*, \ \hat{\rho}(\pi, k + \delta) - \hat{s}\,t_{\alpha/2}^*\right]\right) \to 1 - \alpha.$$

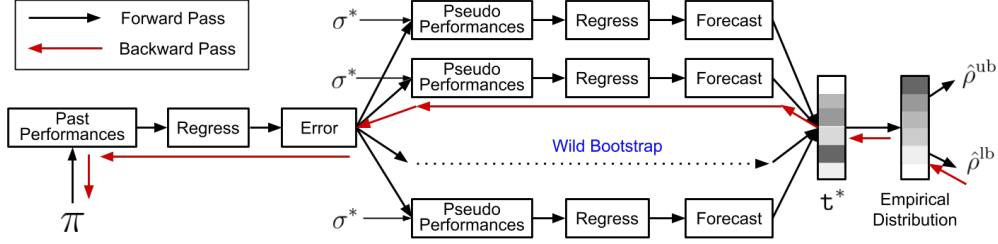

Figure 3: To search for a candidate policy $\pi_c$, regression is first used to analyze the trend of a given policy's past performances. Wild bootstrap then leverages Rademacher variables $\sigma^*$ and the errors from regression to create pseudo-performances. Based on these pseudo-performances, an empirical distribution of the pseudo $\mathtt{t}$-statistic, $\mathtt{t}^*$, of the estimate of future performance, is obtained. The candidate policy $\pi_c$ is found using a differentiation based optimization procedure that maximizes the lower bound, $\hat{\rho}^{\mathrm{lb}}$, computed using the empirical distribution of $\mathtt{t}^*$.

**Remark:** We considered several factors when choosing the wild bootstrap to create pseudo-samples of $\hat{\rho}(\pi, k + \delta)$: (a) because of the time-series structure, there exists no joint distribution between the deterministic sequence of time indices, $X$, and the stochastic performance estimates, $Y$, (b) trajectories from only a *single* sequence of $\{M_i\}_{i=1}^k$ are observed, (c) trajectories could have been generated using different $\beta_i$'s leading to *heteroscedasticity* in the performance estimates $\{\hat{\rho}(\pi, i)\}_{i=1}^k$, (d) different policies $\pi$ can lead to different distributions of performance estimates, even for the same behavior policy $\beta$, and (e) even for a fixed $\pi$ and $\beta$, performance estimates $\{\hat{\rho}(\pi, i)\}_{i=1}^k$ can exhibit heteroskedasticty due to inherent stochasticity in $\{M_i\}_{i=1}^k$ as mentioned in Assumption 1. These factors make popular approaches like pairs bootstrap, residual bootstrap, and block bootstrap not suitable for our purpose. In contrast, the wild bootstrap can take all these factors into account. More discussion on other approaches is available in Appendix D.2.

## 6  Implementation Details

Notice that as the CI $[\hat{\rho}^{\mathrm{lb}}(\pi), \hat{\rho}^{\mathrm{ub}}(\pi)]$ obtained from $\mathscr{C}$ is based on the wild bootstrap procedure, a gradient based optimization procedure for maximizing the lower bound $\hat{\rho}^{\mathrm{lb}}(\pi)$ would require differentiating through the entire bootstrap process. Figure 3 illustrates the high-level steps in this optimization process. More elaborate details and complete algorithms are deferred to Appendix E.

Further, notice that a smaller amount of data results in greater uncertainty and thus wider CIs. While a tighter CI during candidate policy search can be obtained by combining all the past $\mathcal{D}_{\mathrm{train}}$ to increase the amount of data, each safety test should ideally be independent of all the previous tests, and should therefore use data that has never been used before. While it is possible to do so, using only new data for each safety test would be data-inefficient.

To make our algorithm more data efficient, similar to the approach of Thomas et al. [61], we re-use the test data in subsequent tests. As illustrated by the black dashed arrows in Figure 2, this modification introduces a subtle source of error because the data used in consecutive tests are not completely independent. However, the practical advantage of this approach in terms of tighter confidence intervals can be significant. Further, as we demonstrate empirically, the error introduced by re-using test data can be negligible in comparison to the error due to the false assumption of stationarity.

## 7  Empirical Analysis

In this section, we provide an empirical analysis on two domains inspired by safety-critical real-world problems that exhibit non-stationarity. In the following, we first briefly discuss these domains, and in Figure 4 we present a summary of results for eight settings (four for each domain). A more detailed description of the domains and the experimental setup is available in Appendix F.

**Non-Stationary Recommender System (RecoSys):** In this domain, a synthetic recommender system interacts with a user whose interests in different products change over time. Specifically, the reward for recommending each product varies in a seasonal cycle. Such a scenario is ubiquitous in

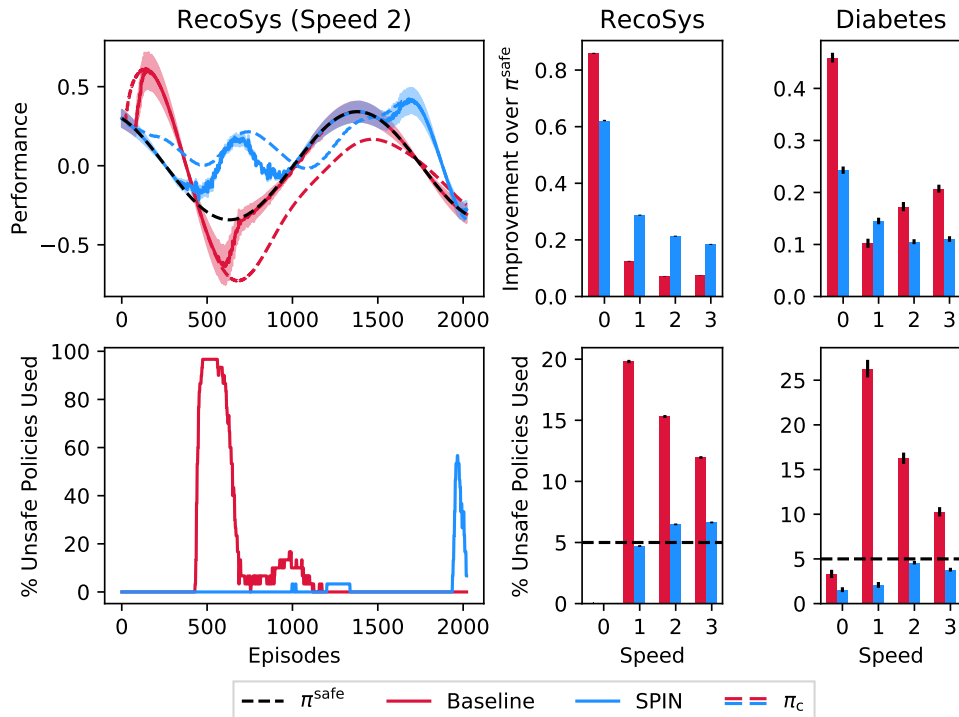

Figure 4: (Top-left) An illustration of a typical learning curve. Notice that SPIN updates a policy whenever there is room for a significant improvement. (Middle and Right) As our main goal is to ensure safety, *while being robust to how a user of our algorithm sets the hyper-parameters (HPs)*, we do *not* show results from the best HP. This choice is motivated by the fact that best performances can often be misleading as it only shows what an algorithm *can* achieve and not what it is *likely* to achieve [32]. Therefore, we present the aggregated results averaged over the *entire sweep* of 1000 HPs per algorithm, per speed, per domain. Shaded regions and intervals correspond to the standard error. See Appendix F.3 and F.4 for more details on these plots and on the data aggregation process.

industrial applications, and updates to an existing system should be made responsibly; if it is not ensured that the new system is better than the existing one, then it might result in a loss of revenue.

**Non-Stationary Diabetes Treatment:** This environment is based on an open-source implementation [68] of the FDA approved Type-1 Diabetes Mellitus simulator (T1DMS) [44, 37] for treatment of Type-1 diabetes. Each episode consists of a day (1440 timesteps–one for each minute in a day) in an *in-silico* patient's life and the blood-glucose level increases when a patient consumes a meal. Too high or too low blood-glucose can lead to fatal conditions like hyperglycemia and hypoglycemia, respectively, and an insulin dosage is therefore required to minimize the risk associated with these conditions. While a doctor's initial dosage prescription is often available, the insulin sensitivity of a patient's internal body organs varies over time, inducing non-stationarity that should be accounted for. The goal of the system is to responsibly update the doctor's initial prescription, ensuring that the treatment is only made better.

The diabetes treatment problem is particularly challenging as the performance trend of policies in this domain can violate Assumption 1 (see Appendix F.1 for more discussion). This presents a more realistic challenge as every policy's performance trend for real-world problems cannot be expected to follow *any* specific trend *exactly*–one can only hope to obtain a coarse approximation of the trend.

For both the domains, (a) we set $\pi^{\text{safe}}$ to a near-optimal policy for the starting MDP $M_1$, representing how a doctor would have set the treatment initially, or how an expert would have set the recommendations, (b) we set the safety level $(1 - \alpha)$ to 95%, (c) we modulate the "speed" of non-stationarity, such that higher speeds represent a faster rate of non-stationarity and a speed of zero represents a stationary domain, and (d) we consider the following two algorithms for comparison: (i) **SPIN:** The proposed algorithm that ensures safety while taking into account the impact of non-stationarity, and (ii) **Baseline:** An algorithm similar to those used in prior works [57, 61, 45], which is aimed at ensuring safety but ignores the impact of non-stationarity (see Appendix F.2 for details).

| | | train-test | 0 | 1 | 2 | 3 | 0 | 1 | 2 | 3 |
|---|---|---|---|---|---|---|---|---|---|---|
| (i) | SPIN | 75%–25% | .56 | .22 | .17 | .14 | 0.0 | 3.6 | 5.1 | 5.4 |
| (ii) | SPIN | 25%–75% | .48 | .29 | .21 | .19 | 0.0 | 4.6 | 6.5 | 7.0 |
| (iii) | SPIN | 50%–50% | .62 | .28 | .21 | .18 | 0.0 | 4.7 | 6.4 | 6.6 |
| (iv) | SPIN-mean | 50%–50% | .70 | .28 | .24 | .19 | 0.2 | 4.9 | 6.3 | 7.1 |
| (v) | NS + No safety | 100%–0% | .73 | .22 | .16 | .19 | 9.4 | 37.6 | 40.2 | 38.6 |
| (vi) | Stationary + Safety | 50%–50% | .85 | .12 | .07 | .07 | 0.0 | 19.8 | 15.3 | 11.9 |

Table 1: Ablation study on the RecoSys domain. (Left) Algorithm. (Middle) Improvement over $\pi^{\text{safe}}$. (Right) Safety violation percentage. Rows (iii) and (vi) correspond to results in Figure 4.

**Results:** An ideal algorithm should adhere to the safety constraint in (1), maximize future performance, and also be robust to hyper-parameters even in the presence of non-stationarity. Therefore, to analyse an algorithm's behavior, we aim to investigate the following three questions:

**Q1:** *How often does an algorithm violate the safety constraint $\rho(\texttt{alg}(\mathcal{D}), k + \delta) \geq \rho(\pi^{safe}, k + \delta)$?* We present these results for SPIN and Baseline on both the domains in Figure 4 (bottom). Baseline ensures safety for the stationary setting (speed $= 0$) but has a severe failure rate otherwise. Perhaps counter-intuitively, the failure rate for Baseline is much *higher* than $5\%$ for *slower* speeds. This can be attributed to the fact that at higher speeds, greater reward fluctuations result in more variance in the performance estimates, causing the CIs within Baseline to be looser, and thereby causing Baseline to have insufficient confidence of policy improvement to make a policy update. Thus, at higher speeds Baseline becomes safer as it reverts to $\pi^{\text{safe}}$ more often. This calls into question the popular misconception that the stationarity assumption is not severe when changes are slow, as in practice slower changes might be harder for an algorithm to identify, and thus might jeopardize safety. By comparison, even though bootstrap CIs do not have guaranteed coverage when using a finite number of samples [24], it still allows SPIN to maintain a failure rate near the $5\%$ target.

**Q2:** *What is the performance gain of an algorithm over the existing known safe policy $\pi^{safe}$?* Notice that any algorithm $\texttt{alg}$ can satisfy the safety constraint in (1) by *never* updating the existing policy $\pi^{\text{safe}}$. Such an $\texttt{alg}$ is not ideal as it will provide no performance gain over $\pi^{\text{safe}}$. In the stationary settings, Baseline provides better performance gain than SPIN while maintaining the desired failure rate. However, in the non-stationary setting, the performance gain of SPIN is higher for the recommender system. For diabetes treatment, both the methods provide similar performance gain but only SPIN does so while being safe (see the bottom-right of Figure 4). The similar performance of Baseline to SPIN despite being unsafe can be attributed to occasionally deploying better policies than SPIN, but having this improvement negated by deploying policies worse than the safety policy (e.g., see the top-left of Figure 4).

**Q3:** *How robust is SPIN to hyper-parameter choices?* To analyze the robustness of our method to the choice of relative train-test data set sizes, the objective for the candidate policy search, and to quantify the benefits of the proposed safety test, we provide an ablation study on the RecoSys domain, for all speeds $(0, 1, 2, 3)$ in Table 1. All other experimental details are the same as in Appendix E.3, except for (iv), where mean performance, as opposed to the lower bound, is optimized during the candidate search. Table 1 shows that the safety violation rate of SPIN is robust to such hyper-parameter changes. However, it is worth noting that too small a test set can make it harder to pass the safety-test, and so performance improvement is small in (i). In contrast, if the proposed safety check procedure for a policy's performance on a non-stationary MDP is removed, then the results can be catastrophic, as can be seen in (v).

# 8 Conclusion

In this paper, we took several first steps towards ensuring safe policy improvement for NS-MDPs. We discussed the difficulty of this problem and presented an algorithm for ensuring the safety constraint in (1) under the assumption of a smooth performance trend. Further, our experimental results call into question the popular misconception that the stationarity assumption is not severe when changes are slow. In fact, it can be quite the opposite: Slow changes can be more *deceptive* and can make existing algorithms, which do not account for non-stationarity, more susceptible to deploying unsafe policies.

# 9    Acknowledgement

We are thankful to Prof. James MacKinnon for sharing valuable insights and other useful references for the wild bootstrap technique. We are also thankful to Shiv Shankar and the anonymous reviewers for providing feedback that helped improve the paper.

This work was supported in part by NSF Award #2018372 and gifts from Adobe Research. Further, this work was also supported in part by NSERC and CIFAR, particularly through funding the Alberta Machine Intelligence Institute (Amii) and the CCAI Chair program.

Research reported in this paper was also sponsored in part by the CCDC Army Research Laboratory under Cooperative Agreement W911NF-17-2-0196 (ARL IoBT CRA). The views and conclusions contained in this document are those of the authors and should not be interpreted as representing the official policies, either expressed or implied, of the Army Research Laboratory or the U.S. Government. The U.S. Government is authorized to reproduce and distribute reprints for Government purposes notwithstanding any copyright notation herein.

# 10    Broader Impact

**Applications:**    We hope that our work brings more attention to the understudied challenge of ensuring safety that is critical for the responsible application of RL algorithms to real-world non-stationary problems. For example, researchers have proposed the use of reinforcement learning algorithms for several medical support systems, ranging from diabetes management [5], to epilepsy [46], to sepsis treatment [50]. These problems involve sequential decision-making, where autonomous systems can improve upon a doctor's prescribed policy by adapting to the non-stationary dynamics of the human body as more data becomes available. In fact, almost all human-computer interaction systems (medical treatment, tutorial recommendations, advertisement marketing, etc.) have a common non-stationary component: humans. Also, in all these use-cases, it is important to ensure that the updates are safe. That is, the updated system should not lead to undesirable financial/medical conditions and should only improve upon the existing policy (e.g., doctor's initial prescription).

**Ethical concerns:**    The proposed method is focused towards ensuring *safety*, defined in terms of the performance of a system. The proposed algorithm to do so makes use of data generated by interacting with a non-stationary MDP. As discussed above, in many cases, non-stationary MDPs are associated with human beings. This raises additional issue of *safety* concerning data privacy and security. The proposed method *does not* resolve any of these issues, and therefore additional care should be taken for adequate data management.

**Note to a wider audience:**    The proposed method relies upon smoothness assumptions that need not be applicable to all problems of interests. For example, when there are jumps or breaks in the time series, then the behavior of the proposed method is not ensured to be safe. Our method also makes use of importance sampling which requires access to the probabilities of the past actions taken under the behavior policy $\beta$. If these probabilities are not available and are instead estimated from data then it may introduce bias and may result in a greater violation of the safety constraint. Further, all of our experiments were conducted on simulated domains, where the exact nature of non-stationarity may *not* reflect the non-stationarity observed during actual interactions in the physical world. Developing simulators that closely mimic the physical world, without incorporating systematic and racial bias, remains an open problem and is complementary to our research. Hence, caution is warranted while quoting results from these simulation experiments.

**Future research directions:**    There are several exciting directions for future research. We used the ordinary importance sampling procedure to estimate past performances of a policy. However, it suffers from high variance and leveraging better importance sampling procedures [31, 56] can be directly beneficial to obtain better estimates of past performances. Leveraging time-series models like ARIMA [14] and their associated wild-bootstrap methods [27, 22, 25] can be a fruitful direction for extending our algorithm to more general settings that have correlated noises or where the performance trend, both locally and globally, can be better modeled using auto-regressive functions. Further, goodness-of-fit tests [15] could be used to search for a time-series model that best fits the application.

## Footnotes

[1]While in machine learning the '*' symbol is often used to denote optimal variables, to be consistent with the bootstrap literature our usage of this symbol denotes pseudo-variables.

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
