[Supplementary Material]

# Towards Safe Policy Improvement for Non-Stationary MDPs (Supplementary Material)

## A  Notation

| Symbol | Meaning |
|---|---|
| $M_i$ | MDP for episode $i$. |
| $\mathcal{S}$ | State set. |
| $\mathcal{A}$ | Action set. |
| $\mathcal{P}_i$ | Transition dynamics for $M_i$. |
| $\mathcal{R}_i$ | Reward function for $M_i$. |
| $\gamma$ | Discounting factor. |
| $d_0$ | Starting state distribution. |
| $\pi$ | Policy. |
| $\pi^{\text{safe}}$ | Given baseline safe policy. |
| $\pi_c$ | A candidate policy that can possibly be used for policy improvement. |
| $\beta_i$ | Behavior policy used to collect data for episode $i$. |
| $G(\pi, m)$ | Discounted episodic return of $\pi$ for MDP $m$. |
| $\rho(\pi, m)$ | Expected discounted episodic return of $\pi$ for MDP $m$. |
| $\rho(\pi, i)$ | Expected discounted episodic return for episode $i$. |
| $\hat{\rho}(\pi, i)$ | An estimate of $\rho(\pi, i)$. |
| $\hat{\rho}^{\text{lb}}(\pi)$ | Lower bound on the future performance of $\pi$. |
| $\hat{\rho}^{\text{ub}}(\pi)$ | Upper bound on the future performance of $\pi$. |
| $k$ | Current episode number. |
| $\delta$ | Number of episodes into the future. |
| $H_i$ | Trajectory during episode $i$. |
| $\mathcal{D}$ | Set of trajectories. |
| $\mathcal{D}_{\text{train}}$ | Partition of $\mathcal{D}$ used for searching $\pi_c$. |
| $\mathcal{D}_{\text{test}}$ | Partition of $\mathcal{D}$ used for safety test. |
| $\texttt{alg}$ | An algorithm. |
| $\alpha$ | Quantity to define the desired safety level $1 - \alpha$. |
| $X$ | Time indices for time-series. |
| $Y$ | Time series values corresponding to $X$. |
| $\hat{Y}$ | Estimates for $Y$. |
| $\phi$ | Basis function for time series forecasting. |
| $\Phi$ | Matrix containing basis for different episode numbers. |
| $w$ | parameters for time series forecasting. |
| $\xi$ | Noise in the observed performances. |
| $\hat{\xi}$ | Estimate for $\xi$. |
| $\hat{\Omega}$ | Diagonal matrix containing $\hat{\xi}^2$. |
| $\hat{s}$ | Standard deviation of the forecast. |
| $\texttt{t}$ | $\texttt{t}-$statistic for the forecast. |
| $\texttt{t}_\alpha$ | $\alpha-$quantile of the $\texttt{t}$ distribution. |
| $\mathscr{C}$ | Function to obtain confidence interval on future performance (using wild bootstrap). |
| $\sigma^*$ | Rademacher random variable. |
| $Y^*$ | Pseudo-variable for $Y$. |
| $\hat{Y}^*$ | Pseudo-variable for $\hat{Y}$. |
| $\xi^*$ | Pseudo-variable for $\xi$. |
| $\hat{\xi}^*$ | Pseudo-variable for $\hat{\xi}$. |
| $\texttt{t}^*$ | Pseudo-variable for $\texttt{t}$. |
| $\texttt{t}^*_\alpha$ | Pseudo-variable for $\texttt{t}_\alpha$. |
| $\hat{s}^*$ | Pseudo-variable for $\hat{s}$. |

Table 2: List of symbols used in the main paper and their associated meanings.

# B  Hardness Results

Several works in the past have presented performance bounds for a policy when executed on an approximated stationary MDP [63, 33, 35, 49, 47, 2]. See Section 6 by Bertsekas and Tsitsiklis [7] for a textbook reference. The technique of our proof for Theorem 1 regarding non-stationary MDPs is based on these earlier results.

**Theorem 1** (Lipschitz smooth performance). *If $\exists \epsilon_P \in \mathbb{R}$ and $\exists \epsilon_R \in \mathbb{R}$ such that for any $M_k$ and $M_{k+1}$, $\forall s \in \mathcal{S}, \forall a \in \mathcal{A}$, $\|\mathcal{P}_k(\cdot|s,a) - \mathcal{P}_{k+1}(\cdot|s,a)\|_1 \leq \epsilon_P$ and $|\mathbb{E}[\mathcal{R}_k(s,a)] - \mathbb{E}[\mathcal{R}_{k+1}(s,a)]| \leq \epsilon_R$, then the performance of any policy $\pi$ is Lipschitz smooth over time, with Lipschitz constant $L := \left( \frac{\gamma R_{max}}{(1-\gamma)^2} \epsilon_P + \frac{1}{1-\gamma} \epsilon_R \right)$. That is,*

$$\forall k \in \mathbb{N}_{>0}, \forall \delta \in \mathbb{N}_{>0}, \quad |\rho(\pi, k) - \rho(\pi, k+\delta)| \leq L\delta. \tag{2}$$

*Proof.* We begin by noting that,

$$|\rho(\pi, k) - \rho(\pi, k+\delta)| \leq \sup_{M_k \in \mathcal{M}, M_{k+\delta} \in \mathcal{M}} |\rho(\pi, M_k) - \rho(\pi, M_{k+\delta})|. \tag{3}$$

We now aim at bounding $|\rho(\pi, M_k) - \rho(\pi, M_{k+\delta})|$ in (3). Let $R_k(s,a) = \mathbb{E}[\mathcal{R}_k(s,a)]$, then notice that the on-policy distribution and the performance of a policy $\pi$ in the episode $k$ can be written as,

$$d^\pi(s, M_k) = (1-\gamma) \sum_{t=0}^{\infty} \gamma^t \Pr(S_t = s | \pi, M_k),$$

$$\rho(\pi, M_k) = (1-\gamma)^{-1} \sum_{s \in \mathcal{S}} d^\pi(s, M_k) \sum_{a \in \mathcal{A}} \pi(a|s) R_k(s,a).$$

We begin the proof by expanding the absolute difference between the two performances as follows:

$$|\rho(\pi, M_k) - \rho(\pi, M_{k+\delta})|$$
$$= |\rho(\pi, M_k) - \rho(\pi, M_{k+1}) + \rho(\pi, M_{k+1}) - ... - \rho(\pi, M_{k+\delta-1}) + \rho(\pi, M_{k+\delta-1}) - \rho(\pi, M_{k+\delta})|$$

$$\leq \sum_{i=k}^{k+\delta-1} |\rho(\pi, M_i) - \rho(\pi, M_{i+1})|. \tag{4}$$

To simplify further, we introduce a temporary notation $\Delta(s,a) := R_i(s,a) - R_{i+1}(s,a)$. Now on expanding each of the consecutive differences in (4) and multiplying by $(1-\gamma)$ on both sides:

$$(1-\gamma)|\rho(\pi, M_i) - \rho(\pi, M_{i+1})|$$

$$= \left| \sum_{s \in \mathcal{S}} d^\pi(s, M_i) \sum_{a \in \mathcal{A}} \pi(a|s) R_i(s,a) - \sum_{s \in \mathcal{S}} d^\pi(s, M_{i+1}) \sum_{a \in \mathcal{A}} \pi(a|s) R_{i+1}(s,a) \right|$$

$$= \left| \sum_{s \in \mathcal{S}} \sum_{a \in \mathcal{A}} \pi(a|s) \Big( d^\pi(s, M_i) R_i(s,a) - d^\pi(s, M_{i+1}) R_{i+1}(s,a) \Big) \right|$$

$$= \left| \sum_{s \in \mathcal{S}} \sum_{a \in \mathcal{A}} \pi(a|s) \Big( d^\pi(s, M_i)(R_{i+1}(s,a) + \Delta(s,a)) - d^\pi(s, M_{i+1}) R_{i+1}(s,a) \Big) \right|$$

$$= \left| \sum_{s \in \mathcal{S}} \sum_{a \in \mathcal{A}} \pi(a|s) \Big( d^\pi(s, M_i) - d^\pi(s, M_{i+1}) \Big) R_{i+1}(s,a) + \sum_{s \in \mathcal{S}} \sum_{a \in \mathcal{A}} \pi(a|s) d^\pi(s, M_i) \Delta(s,a) \right| \tag{5}$$

In the following, we bound the terms in (5) using the following three steps, (a) use Cauchy Schwartz inequality and bound each possible negative term with its absolute value, (b) bound each reward $R_{i+1}(s,a)$ using $R_{max}$ and use the Lipschitz smoothness assumption to bound each $\Delta(s,a)$ using $\epsilon_R$, and (c) equate sum of probabilities to one. Formally,

$$(1-\gamma)|\rho(\pi, M_i) - \rho(\pi, M_{i+1})|$$

$$\overset{(a)}{\leq} \sum_{s\in\mathcal{S}}\sum_{a\in\mathcal{A}} \pi(a|s)\,|d^\pi(s, M_i) - d^\pi(s, M_{i+1})|\,|R_{i+1}(s, a)| + \sum_{s\in\mathcal{S}}\sum_{a\in\mathcal{A}} \pi(a|s)d^\pi(s, M_i)\,|\Delta(s, a)|$$

$$\overset{(b)}{\leq} R_{\max}\sum_{s\in\mathcal{S}}\sum_{a\in\mathcal{A}} \pi(a|s)\,|d^\pi(s, M_i) - d^\pi(s, M_{i+1})| + \epsilon_R \sum_{s\in\mathcal{S}}\sum_{a\in\mathcal{A}} \pi(a|s)d^\pi(s, M_i)$$

$$\overset{(c)}{=} R_{\max}\sum_{s\in\mathcal{S}} |d^\pi(s, M_i) - d^\pi(s, M_{i+1})| + \epsilon_R. \tag{6}$$

To simplify (6) further, we make use of the following property,

**Property 1** (Achiam et al. [2]). *Let $P_i^\pi \in \mathbb{R}^{|\mathcal{S}|\times|\mathcal{S}|}$ be the transition matrix ($s'$ in rows and $s$ in columns) resulting due to $\pi$ and $P_i$, i.e., $\forall t$, $P_i^\pi(s', s) \coloneqq \Pr(S_{t+1} = s'|S_t = s, \pi, M_i)$, and let $d^\pi(\cdot, M_i) \in \mathbb{R}^{|\mathcal{S}|}$ denote the vector of probabilities for each state, then[2]*

$$\sum_{s\in\mathcal{S}} |d^\pi(s, M_i) - d^\pi(s, M_{i+1})| \leq \gamma(1-\gamma)^{-1} \left\| (P_i^\pi - P_{i+1}^\pi)d^\pi(\cdot, M_i) \right\|_1.$$

Using Property 1,

$$\sum_{s\in\mathcal{S}} |d^\pi(s, M_i) - d^\pi(s, M_{i+1})|$$

$$\overset{(d)}{\leq} \gamma(1-\gamma)^{-1} \sum_{s'\in\mathcal{S}} \left| \sum_{s\in\mathcal{S}} \left( P_i^\pi(s', s) - P_{i+1}^\pi(s', s) \right) d^\pi(s, M_i) \right|$$

$$\leq \gamma(1-\gamma)^{-1} \sum_{s'\in\mathcal{S}}\sum_{s\in\mathcal{S}} \left| P_i^\pi(s', s) - P_{i+1}^\pi(s', s) \right| d^\pi(s, M_i)$$

$$= \gamma(1-\gamma)^{-1} \sum_{s'\in\mathcal{S}}\sum_{s\in\mathcal{S}} \left| \sum_{a\in\mathcal{A}} \pi(a|s)\Big( \Pr(s'|s, a, M_i) - \Pr(s'|s, a, M_{i+1}) \Big) \right| d^\pi(s, M_i)$$

$$\leq \gamma(1-\gamma)^{-1} \sum_{s'\in\mathcal{S}}\sum_{s\in\mathcal{S}}\sum_{a\in\mathcal{A}} \pi(a|s)\,|\Pr(s'|s, a, M_i) - \Pr(s'|s, a, M_{i+1})|\, d^\pi(s, M_i)$$

$$= \gamma(1-\gamma)^{-1} \sum_{s\in\mathcal{S}}\sum_{a\in\mathcal{A}} \pi(a|s)d^\pi(s, M_i) \sum_{s'\in\mathcal{S}} |\Pr(s'|s, a, M_i) - \Pr(s'|s, a, M_{i+1})|$$

$$\overset{(e)}{\leq} \gamma(1-\gamma)^{-1} \sum_{s\in\mathcal{S}}\sum_{a\in\mathcal{A}} \pi(a|s)d^\pi(s, M_i)\epsilon_P$$

$$= \gamma(1-\gamma)^{-1}\epsilon_P, \tag{7}$$

where (d) follows from expanding the L1 norm of a matrix-vector product, and (e) follows from using the Lipschitz smoothness to bound the difference between successive transition matrices. Combining (6) and (7),

$$|\rho(\pi, M_i) - \rho(\pi, M_{i+1})| \leq (1-\gamma)^{-1} \left( R_{\max}\gamma(1-\gamma)^{-1}\epsilon_P + \epsilon_R \right).$$

$$= \frac{\gamma R_{\max}}{(1-\gamma)^2}\epsilon_P + \frac{1}{1-\gamma}\epsilon_R. \tag{8}$$

Finally, combining (4) and (8),

$$|\rho(\pi, M_i) - \rho(\pi, M_{i+\delta})| \leq \sum_{i=k}^{k+\delta-1} \left( \frac{\gamma R_{\max}}{(1-\gamma)^2}\epsilon_P + \frac{1}{1-\gamma}\epsilon_R \right)$$

$$= \delta \left( \frac{\gamma R_{\max}}{(1-\gamma)^2}\epsilon_P + \frac{1}{1-\gamma}\epsilon_R \right).$$

$\square$

**Tightness of the bound:** In this paragraph, we present an NS-MDP where (2) holds with exact equality, illustrating that the bound given by Theorem 1 is tight.

Consider the NS-MDP given in Figure 5. Let $\gamma = 0$ and let $\mathcal{A} = \{a\}$ such that the size of action set $|\mathcal{A}| = 1$. Let the state set $\mathcal{S} = \{s_1, s_2\}$ and let the initial state for an episode always be state $s_1$. Let rewards be in the range $[-1, +1]$ such that $R_{\max} = 1$.

Figure 5: Example NS-MDP.

Notice that for NS-MDP in Figure 5, $\epsilon_R = |\mathbb{E}[\mathcal{R}_1(s_1, a)] - \mathbb{E}[\mathcal{R}_2(s_1, a)]| = 0.2$ as

$$R_1(s_1, a) = \mathbb{E}[\mathcal{R}_1(s_1, a)] = 1 \cdot (+1) + 0 \cdot (-1) = 1,$$
$$R_2(s_1, a) = \mathbb{E}[\mathcal{R}_2(s_1, a)] = 0.9 \cdot (+1) + 0.1 \cdot (-1) = 0.8.$$

Similarly,

$$\epsilon_P = |\mathcal{P}_1(s_1|s_1, a) - \mathcal{P}_2(s_1|s_1, a)| + |\mathcal{P}_1(s_2|s_1, a) - \mathcal{P}_2(s_2|s_1, a)| = 0.2.$$

Therefore, substituting the values $\gamma = 0, R_{\max} = 1, \epsilon_P = \epsilon_R = 0.2$, and $\delta = 1$ in (2), we get

$$|\rho(\pi, M_1) - \rho(\pi, M_2)| \leq 0.2. \tag{9}$$

Now to illustrate that the bound is tight, we compute the true difference in performances of a policy $\pi$ for the MDPs given in Figure 5, i.e., the LHS of (9). Notice that

$$\rho(\pi, M_1) = (1-\gamma)^{-1} \sum_{s \in \mathcal{S}} d^\pi(s, M_1) \sum_{a \in \mathcal{A}} \pi(a|s)R_1(s, a) \stackrel{\text{(a)}}{=} R_1(s_1, a) = 1,$$

where **(a)** follows because (i) $\gamma = 0$, (ii) as there is only a single action, $\pi(a|s) = 1$, and (iii) since $s_1$ is the starting state and $\gamma = 0$, therefore, $d^\pi(s_1, M_1) = 1$ and $d^\pi(s_2, M_1) = 0$. Similarly, $\rho(\pi, M_2) = R_2(s_1, a) = 0.8$. Therefore, $|\rho(\pi, M_1) - \rho(\pi, M_2)| = 0.2$, which is exactly equal to the value of bound in (9).

## C  Uncertainty Estimation

Our Theorem 2 makes use of a property proven by Djogbenou et al. [23]. This property by Djogbenou et al. [23] was established for inference about the *parameters* of a regression model. We leverage this property to obtain confidence intervals for *predictions* of future performance. In the following section, we first review their results and then in the section thereafter we present the proof of Theorem 2.

### C.1  Preliminary

Before moving forward, we first revisit all the necessary notations and review the result by Djogbenou et al. [23]. For a regression problem, let $Y \in \mathbb{R}^{k \times 1}$ be the stochastic observations, let $\Phi \in \mathbb{R}^{k \times d}$

be the deterministic predicates, and let $w \in \mathbb{R}^{d \times 1}$ be the regression parameters. Let $\xi \in \mathbb{R}^{k \times 1}$ be a vector of $k$ independent noises. The linear system of equations for regression is then given by,

$$Y = \Phi w + \xi. \tag{10}$$

The least-squares estimate $\hat{w}$ of $w$ is given by $\hat{w} := (\Phi^\top \Phi)^{-1} \Phi^\top Y$ and the estimate $\hat{Y} := \Phi \hat{w}$. Subsequently, co-variance of the estimate $\hat{w}$ can be computed as following,

$$
\begin{aligned}
V := \mathbb{V}(\hat{w}) =& \mathbb{E}\left[ (\hat{w} - \mathbb{E}[\hat{w}]) \, (\hat{w} - \mathbb{E}[\hat{w}])^\top \right] \\
=& \mathbb{E}\left[ \left( (\Phi^\top \Phi)^{-1} \Phi^\top (Y - \mathbb{E}[Y]) \right) \left( (\Phi^\top \Phi)^{-1} \Phi^\top (Y - \mathbb{E}[Y]) \right)^\top \right] \\
=& \mathbb{E}\left[ \left( (\Phi^\top \Phi)^{-1} \Phi^\top \xi \right) \left( (\Phi^\top \Phi)^{-1} \Phi^\top \xi \right)^\top \right] \\
=& \mathbb{E}\left[ (\Phi^\top \Phi)^{-1} \Phi^\top \xi \xi^\top \Phi (\Phi^\top \Phi)^{-1} \right] \\
\overset{\textbf{(a)}}{=}& (\Phi^\top \Phi)^{-1} \Phi^\top \mathbb{E}\left[ \xi \xi^\top \right] \Phi (\Phi^\top \Phi)^{-1} \\
=& (\Phi^\top \Phi)^{-1} \Phi^\top \Omega \Phi (\Phi^\top \Phi)^{-1},
\end{aligned}
\tag{11}
$$

where **(a)** follows from the fact that $\Phi$ is deterministic, and $\Omega$ is the co-variance matrix of the mean-zero and heteroscedastic noises $\xi$. Notice that as the noises are independent, the off-diagonal terms in $\Omega$ are zero. However, since the true $\Omega$ is not known, it can be estimated using $\hat{\Omega}$ which contains the squared errors from the OLS estimate [42]. That is, let $\hat{\xi} := \hat{Y} - Y$, then $\hat{\Omega}$ is a diagonal matrix with $\hat{\xi}^2$ in the diagonal. Let such an estimator of $\mathbb{V}(w)$ be,

$$\hat{V} := (\Phi^\top \Phi)^{-1} \Phi^\top \hat{\Omega} \Phi (\Phi^\top \Phi)^{-1}. \tag{12}$$

Let $b^\top w$ be a desired null hypothesis with $b^\top b = 1$. Let $\mathtt{t}_b$, the $\mathtt{t}$-statistic for testing this hypothesis, and its pseudo-sample $\mathtt{t}_b^*$ obtained using the wild bootstrap procedure with Rademacher variables $\sigma^*$ be (see Section 5 in the main body for exact steps),

$$\mathtt{t}_b = \frac{b^\top (\hat{w} - w)}{\sqrt{b^\top \hat{V} b}}, \qquad \mathtt{t}_b^* := \frac{b^\top (\hat{w}^* - \hat{w})}{\sqrt{b^\top \hat{V}^* b}}. \tag{13}$$

Note that in (13), the subscript of $b$ is *not* related to percentile of the previously defined $\mathtt{t}$-statistic: $\mathtt{t}$ in the main paper. $\mathtt{t}_b$ and $\mathtt{t}_b^*$ are new variables.

Now we state the result we use from the work by Djogbenou et al. [23]. This result require two main assumptions. Our presentations of these assumptions are slightly different from the exact statements given by Djogbenou et al. [23]. The differences are (a) we make the assumptions tighter than what is required for their results to hold, and (b) we ignore a third assumption that is related to cluster sizes, as our setting is a special case where the cluster size is equal to $1$. We call these assumptions *requirements* to distinguish them from our assumptions.

**Requirement 1** (Independence). $\forall i \in [1, k]$, *the noise terms $\xi_i$'s are mean-zero, bounded, and independent random variables.*

**Requirement 2** (Positive Definite). $(\Phi^\top \Phi)^{-1}$ *is positive-definite and $\exists C_2 > 0$ such that $\|\Phi\|_\infty < C_2$.*

**Lemma 1** (Theorem 3.2 Djogbenou et al. [23]). *Under Requirements 1 and 2, if $\mathbb{E}[\sigma^{*3}] < \infty$ and if the true value of $w$ is given by (10), then as $k \to \infty$,*

$$\mathrm{Pr}\left( \sup_{x \in \mathbb{R}} |\mathrm{Pr}(\mathtt{t}_b^* < x) - \mathrm{Pr}(\mathtt{t}_b < x)| > \alpha \right) \to 0.$$

## C.2 Proof of Coverage Error

First, we recall the notations established in the main body, which are required for the proof. Using similar steps to those in (11), it can be seen that the variance $V_f$ of the estimator $\hat{\rho}(\pi, k + \delta) := \phi(k + \delta)\hat{w}$ of future performance is

$$V_f = \phi(k + \delta)(\Phi^\top \Phi)^{-1} \Phi^\top \Omega \Phi (\Phi^\top \Phi)^{-1} \phi(k + \delta)^\top.$$

Similar to before, let an estimate $\hat{V}_f$ of $V_f$ be defined as,

$$\hat{V}_f = \phi(k + \delta)(\Phi^\top \Phi)^{-1}\Phi^\top \hat{\Omega}\Phi(\Phi^\top \Phi)^{-1}\phi(k + \delta)^\top, \tag{14}$$

where $\hat{\Omega}$ is the same as in (12). Recall from Section 5 that the sample standard deviation of the $\phi(k + \delta)\hat{w}$ is $\hat{s} = \sqrt{\hat{V}_f}$ and the pseudo standard deviation $\hat{s}^* := \sqrt{\hat{V}_f^*}$, where the pseudo variables are created using wild bootstrap procedure outlined in Section 5. Similarly, recall that the t-statistic and the pseudo t-statistic for estimating future performance are given by

$$\mathtt{t} := \frac{\hat{\rho}(\pi, k + \delta) - \rho(\pi, k + \delta)}{\hat{s}}, \qquad \mathtt{t}^* := \frac{\hat{\rho}(\pi, k + \delta)^* - \hat{\rho}(\pi, k + \delta)}{\hat{s}^*}.$$

For the purpose of Theorem 2, we use a Fourier basis of order $d$, which is given by [9]:

$$\phi(x) := \left\{ \frac{\sin(2\pi nx)}{C} \middle| n \in [1, d] \right\} \cup \left\{ \frac{\cos(2\pi nx)}{C} \middle| n \in [1, d] \right\} \cup \left\{ \frac{1}{C} \right\}, \tag{15}$$

where $C := \sqrt{d + 1}$.

**Theorem 2** (Consistent Coverage). *Under Assumptions 1 and 2, if the set of trajectories $\{H_i\}_{i=1}^k$ are independent and if $\phi(x)$ is a Fourier basis of order $d$, then as $k \to \infty$,*

$$\Pr\left( \rho(\pi, k + \delta) \in \left[ \hat{\rho}(\pi, k + \delta) - \hat{s}t^*_{1-\alpha/2}, \ \hat{\rho}(\pi, k + \delta) - \hat{s}t^*_{\alpha/2} \right] \right) \to 1 - \alpha. \tag{16}$$

*Proof.* For the purpose of this proof, we will make use Lemma 1. Therefore, we first discuss how our method satisfies the requirements for Lemma 1.

To satisfy Requirement 1, recall that in the proposed method, the estimates $\{\hat{\rho}(\pi, i)\}_{i=1}^k$ of past performances are obtained using counter-factual reasoning. Therefore, satisfying Requirement 1 in our method requires consideration of two sources of noise: (a) the noise resulting from the inherent stochasticity in the non-stationary MDP sequence, as given in Assumption 1, and (b) the other noise resulting due to our use of importance sampling to estimate past performances $\{\rho(\pi, i)\}_{i=1}^k$, which are subsequently used to obtain the forecast for $\rho(\pi, k + \delta)$.

Notice that the noises $\{\xi_i\}_{i=1}^k$ inherent to the non-stationary MDP are both mean-zero and independent because of Assumption 1. Further, as importance sampling is unbiased and uses independent draws of trajectories $\{H_i\}_{i=1}^k$, the additional noises in the estimates $\{\hat{\rho}(\pi, i)\}_{i=1}^k$ are also mean-zero and independent. The boundedness condition of each $\xi_i$ also holds as (a) all episodic returns are bounded, which is because every reward is bounded between $[-R_{\max}, R_{\max}]$ and $\gamma < 1$, and (b) following Assumption 2, the denominator of importance sampling ratios are lower bounded by $C$. Therefore, importance weighted returns are upper bounded by a finite constant. This makes the noise from importance sampling estimates also bounded. Hence, all the noises in our performance estimates are independent, bounded, and mean zero.

To satisfy Requirement 2, note that as $\Phi^\top \Phi$ is an inner product matrix, it has to be positive semi-definite. Further, as the Fourier basis creates linearly independent features, when $k > d$ (i.e., it has more samples than number of parameters) the matrix will have full column-rank. Combining these two points it can be seen that $\Phi^\top \Phi$ is a positive-definite matrix and as the eigenvalues of $(\Phi^\top \Phi)^{-1}$ are just the reciprocals of the eigenvalues of $\Phi^\top \Phi$, the matrix $(\Phi^\top \Phi)^{-1}$ is also positive-definite. Second half of Requirement 2 is trivially satisfied as all the values of $\phi(x)$ are in $[-1/C, 1/C]$.

Finally, note that when $\phi : \mathbb{N} \to \mathbb{R}^{1 \times d}$ is a Fourier basis then $\forall x \in \mathbb{R}, \ \phi(x)\phi(x)^\top = 1$. To see why, notice from (15) that

$$\phi(x)\phi(x)^\top = \sum_{n=1}^d \left( \frac{\sin(2\pi nx)}{C} \right)^2 + \sum_{n=1}^d \left( \frac{\cos(2\pi nx)}{C} \right)^2 + \left( \frac{1}{C} \right)^2$$

$$= \frac{\sum_{n=1}^d \left( \sin^2(2\pi nx) + \cos^2(2\pi nx) \right) + 1}{C^2} \stackrel{\text{(a)}}{=} \frac{d + 1}{C^2} = 1, \tag{17}$$

where **(a)** follows from the trignometric inequality that $\forall x \in \mathbb{R} \ \sin^2(x) + \cos^2(x) = 1$.

Now we are ready for the complete proof. For brevity, we define $\mathcal{C} := \left[\hat{\rho}(\pi, k+\delta) - \hat{s}\mathsf{t}^*_{1-\alpha/2}, \; \hat{\rho}(\pi, k+\delta) - \hat{s}\mathsf{t}^*_{\alpha/2}\right]$, $\rho := \rho(\pi, k+\delta)$, and $\hat{\rho} := \hat{\rho}(\pi, k+\delta)$, and expand the LHS of (16),

$$
\begin{aligned}
\Pr\left(\rho \in \mathcal{C}\right) &= \Pr\left(\hat{\rho} - \hat{s}\mathsf{t}^*_{1-\alpha/2} \leq \rho \leq \hat{\rho} - \hat{s}\mathsf{t}^*_{\alpha/2}\right) \\
&= \Pr\left(-\hat{s}\mathsf{t}^*_{1-\alpha/2} \leq \rho - \hat{\rho} \leq -\hat{s}\mathsf{t}^*_{\alpha/2}\right) \\
&= \Pr\left(\hat{s}\mathsf{t}^*_{1-\alpha/2} \geq \hat{\rho} - \rho \geq \hat{s}\mathsf{t}^*_{\alpha/2}\right) \\
&= \Pr\left(\mathsf{t}^*_{1-\alpha/2} \geq \frac{\hat{\rho} - \rho}{\hat{s}} \geq \mathsf{t}^*_{\alpha/2}\right). \\
&= \Pr\left(\mathsf{t}^*_{1-\alpha/2} \geq \mathsf{t} \geq \mathsf{t}^*_{\alpha/2}\right) \\
&= \Pr\left(\mathsf{t} \leq \mathsf{t}^*_{1-\alpha/2}\right) - \Pr\left(\mathsf{t} \leq \mathsf{t}^*_{\alpha/2}\right). \quad (18)
\end{aligned}
$$

To simplify (18), let $b = \phi(k+\delta)^\top$. Under this instantiation of $b$, the null hypothesis $b^\top w$ in C.1 for our setting corresponds to $\phi(k+\delta)w$, which is the true future performance under Assumption 1. Further, for this instantiation of $b$, note from (17) that $b^\top b = 1$. Now, it can be seen from (14) that $\hat{V}_f = b^\top \hat{V} b$, and $\hat{V}_f^* = b^\top \hat{V}^* b$. Thus, $\mathsf{t} = \mathsf{t}_b$ and $\mathsf{t}^* = \mathsf{t}_b^*$. Finally, note that as $\sigma^*$ corresponds to the Rademacher random variable, $\mathbb{E}[\sigma^{*3}] = 0$.

Therefore, leveraging Lemma 1, in the limit, for any $x$, we can substitute $\Pr(\mathsf{t} < x)$ with $\Pr(\mathsf{t}^* < x)$ in (18), This substitution yields.

$$
\begin{aligned}
\Pr\left(\rho \in \mathcal{C}\right) &\to \Pr\left(\mathsf{t}^* \leq \mathsf{t}^*_{1-\alpha/2}\right) - \Pr\left(\mathsf{t}^* \leq \mathsf{t}^*_{\alpha/2}\right) \\
&= (1 - \alpha/2) - (\alpha/2) \\
&= 1 - \alpha. \qquad \qquad \square
\end{aligned}
$$

Notice that using the Fourier basis, we were able to satisfy the condition that $b^\top b = 1$ directly. This allowed us to leverage Lemma 1 without much modification. However, as noted by Djogbenou et al. [23], the constraint on $b^\top b$ is not necessary and was used to simplify the proof.

## D Extended Discussion on Bootstrap

The goal of this section is to provide additional discussion on (wild) bootstrap for completeness. Therefore, this section contains a summary of existing works and has no original technical contribution. We begin by first discussing the idea behind any general bootstrap and the wild bootstrap method. Subsequently, we discuss alternatives to wild bootstrap.

In many practical applications, it is often desirable to infer distributional properties (e.g., CIs) of a desired statistic of data (e.g., mean). However, in practice, it is often not possible to get multiple estimates of the desired statistic in a data-efficient way. To address this problem, bootstrap methods have received wide popularity in the field of computational statistics [24].

The core principle of any bootstrap procedure is to *re-sample* the observed data-set $\mathcal{D}$ and construct multiple *pseudo data-sets* $\mathcal{D}^*$ in a way that closely mimics the original *data generating process* (DGP). This allows to create an *empirical distribution* of the desired statistic by leveraging multiple pseudo data-sets $\mathcal{D}^*$ [24]. For example, an empirical distribution containing $B$ estimates of the sample mean can be obtained by generating $B$ pseudo data-sets, where each data-set contains $N$ samples uniformly drawn (with replacement) from the original data-set of size $N$.

For an excellent introduction to bootstrap CIs, refer to the works by Efron and Tibshirani [24] and DiCiccio and Efron [21]. The book by Hall [29] provides a thorough treatment of these methods using *Edgeworth expansion*, illustrating when and how bootstrap methods can provide significant advantage over other methods. For a very readable practitioner's guide that touches upon several important aspects, refer to the work by Carpenter and Bithell [11].

### D.1 Why does wild bootstrap work?

The original idea of wild bootstrap was proposed by Wu et al. [65] and later developed by Liu et al. [41], Mammen [43], and Davidson and Flachaire [19, 20]. The following summary about the wild bootstrap process is based on an excellent tutorial by MacKinnon [42].

Consider the system of equations in (10). The key idea of wild-bootstrap is that the uncertainty in regression estimates (of parameters/predictions) is due to the noise $\xi$ in the observations. Therefore, if the pseudo-data $Y^*$ is generated such that the noise $\xi^*$ in the data generating process for $Y^*$ resembles the properties of the true underlying noise $\xi$, then with multiple redraws of such $Y^*$ one can obtain an empirical distribution of the desired statistic (which for our case, corresponds to the forecast of a policy $\pi$'s performance). This can then be used to estimate the CIs.

As true noise $\xi$ is unobserved, it raises a question about how to estimate its properties to generate $Y^*$. Fortunately, as ordinary least-squares is an unbiased estimator of parameters/predictions [62], regression errors $\hat{\xi}$ can be used as a substitute for the true noise. Therefore, to mimic the underlying data generating process, it would be ideal to have bootstrap error terms $\xi^*$ that have similar moments as $\hat{\xi}$. Following the work by Davidson and Flachaire [19], we set $Y^* := \hat{Y} + \xi^*$, where $\xi^* := \hat{\xi} \odot \sigma^*$, and $\sigma^* \in \mathbb{R}^{k \times 1}$ is the independent Rademacher random variable (i.e., $\forall i \in [1, k]$, $\Pr(\sigma_i^* = +1) = \Pr(\sigma_i^* = -1) = 0.5$). This choice of $\sigma_i^*$, for all $i \in [1, k]$, ensures that $\xi_i^*$ has the desired zero mean and the same higher-order *even* moments as $\hat{\xi}_i$ because,

$$\forall i, \ \mathbb{E}[\sigma_i^*] = 0, \ \mathbb{E}[\sigma_i^{*2}] = 1, \ \mathbb{E}[\sigma_i^{*3}] = 0, \ \mathbb{E}[\sigma_i^{*4}] = 1.$$

Therefore, for the purpose of this paper, pseudo performances $Y^*$ generated using pseudo-noise $\xi^*$ allow generating a distribution of $\hat{\rho}(\pi, k + \delta)^*$ that closely mimics the distribution of forecasts $\hat{\rho}(\pi, k + \delta)$ that would have been generated if we had the true underlying data generating process.

### D.2 Why not use other bootstrap methods?

One popular non-parametric technique for bootstrapping in regression is to re-sample, with replacement, $(x, y)$ pairs from the set of observed samples $(X, Y)$ [11]. However, in our setup, $X$ variable corresponds to the (deterministic) time index and thus there exists no joint distribution between the $X$ and the $Y$ variables from where time can be sampled stochastically. Therefore, paired re-sampling will not mimic the underlying data generative process in our setting.

A semi-parametric technique overcomes the above problem by only re-sampling $Y$ variable as follows. First, a model is fit to the observed data $(X, Y)$ and predictions $\hat{Y}$ are obtained. Then an empirical cumulative distribution function, $\Psi(e)$ of all the errors, $e := Y - \hat{Y}$, is obtained. Subsequently, new bootstrapped variables are created as $Y^* := \hat{Y} + \xi^*$, where $\xi^*$ is the re-sampled noise from $\Psi(e)$ [24]. However, such a process assumes that noises are homoscedastic, which will be severely violated for our purpose.

Another popular technique for *auto-correlated* data uses the idea of *block re-sampling* [24]. However, this assumes that the underlying process is stationary, and hence is not suitable for our purpose.

### D.3 Why not use standard `t`-test?

Standard `t`-test assumes that the predictions will follow the student-`t` distribution. Such an assumption can be severely violated, specially in the presence of heteroscedasticity, and heavy tailed noises, when the sample size is not sufficiently large. Unfortunately, in our setting, use of multiple behavior policies result in heteroscedasticity and importance sampling results in heavy tailed distribution [59] for counterfactual estimates of past performances.

It can also be shown that for a finite sample of size $n$, the coverage error of CIs obtained using standard `t`-statistic is of order $O(n^{-1/2})$ [62, 29]. In comparison, it can be shown using Edgeworth expansions [29] that the coverage error rate of CIs obtained using bootstrap methods typically provide higher-order refinement by providing error rates up to $O(n^{-p/2})$, where $p \in [1, 3]$ [28, 21, 29]. For more elaborate discussions in the context of wild bootstrap, see the work by Kline and Santos [36] and by Djogbenou et al. [23]. Also, see the work by Mammen [43] for detailed empirical comparison of standard t-test against wild-bootstrap.

## E Algorithm

In Algorithms 1-3,[3] we provide the steps for our method: SPIN. In Algorithm 2, PDIS is shorthand for per-decision importance sampling discussed in Section 5. In the following, we discuss certain aspects of SPIN, especially pertaining to the search of a candidate policy $\pi_c$.

---

**Algorithm 1:** Forecast

1. **Input** Predicates $\Phi$, Targets $Y$, Forecast time(s) $\tau$
2. $H \leftarrow (\Phi^\top \Phi)^{-1} \Phi^\top$
3. $\varphi \leftarrow [\phi(\tau_1), ..., \phi(\tau_\delta)]$
4. $\hat{Y} \leftarrow \Phi H Y$
5. $\hat{\rho} \leftarrow \mathtt{mean}(\varphi H Y)$
6. $\hat{\xi} \leftarrow Y - \hat{Y}$
7. $\hat{\Omega} \leftarrow \mathtt{diag}(\hat{\xi}^2)$
8. $\hat{V} \leftarrow \mathtt{mean}(\varphi H \hat{\Omega} H^\top \varphi^\top)$
9. Return $\hat{\rho}, \hat{V}, \hat{\xi}$

---

**Algorithm 2:** PI: Prediction Interval

1. **Input** Data $\mathcal{D}$, Policy $\pi$, Safety-violation rate $\alpha$, Forecast time(s) $\tau$
2. $\Phi \leftarrow \emptyset, Y \leftarrow \emptyset$

   # Create regression variables
3. **for** $(k, h) \in \mathcal{D}$ **do**
4. $\quad$ $\hat{\rho}(\pi, k) \leftarrow \mathrm{PDIS}(\pi, h)$
5. $\quad$ $\Phi.\mathtt{append}(\phi(k))$
6. $\quad$ $Y.\mathtt{append}(\hat{\rho}(\pi, k))$

7. $\hat{\rho}, \hat{V}, \hat{\xi} \leftarrow \mathrm{Forecast}(\Phi, Y, \tau)$

   # Wild Bootstrap (in parallel)
8. $\mathtt{t}^* \leftarrow \emptyset, \mathtt{t}^{**} \leftarrow \emptyset$
9. **for** $i \in [1, ..., B]$ **do**
10. $\quad$ $\sigma^* \leftarrow [\pm 1, \pm 1, ..., \pm 1]$
11. $\quad$ $\xi^* \leftarrow \hat{\xi} \odot \sigma^*$
12. $\quad$ $Y^* \leftarrow \hat{Y} + \xi^*$
13. $\quad$ $\hat{\rho}^*, \hat{V}^*, \_ \leftarrow \mathrm{Forecast}(\Phi, Y^*, \tau)$
14. $\quad$ $\mathtt{t}^*[i] \leftarrow (\hat{\rho}^* - \hat{\rho})/\sqrt{\hat{V}^*}$

   # Get prediction interval
15. $\mathtt{t}^{**} \leftarrow \mathtt{sort}(\mathtt{t}^*)$
16. $\hat{\rho}^{\mathrm{lb}} \leftarrow \hat{\rho} - \mathtt{t}^{**}[(1 - \alpha/2)B]\sqrt{\hat{V}}$
17. $\hat{\rho}^{\mathrm{ub}} \leftarrow \hat{\rho} - \mathtt{t}^{**}[(\alpha/2)B]\sqrt{\hat{V}}$

18. Return $(\hat{\rho}^{\mathrm{lb}}, \hat{\rho}^{\mathrm{ub}})$

---

**Algorithm 3:** SPIN: Safe Policy Improvement for Non-stationary settings

1. **Input** Safety-violation rate $\alpha$, Initial safe policy $\pi^{\mathrm{safe}}$, Entropy-regularizer $\lambda$, Batch-size $\delta$
2. **Initialize** $\mathcal{D}_{\mathrm{train}} \leftarrow \emptyset, \mathcal{D}_{\mathrm{test}} \leftarrow \emptyset$, $\pi \leftarrow \pi_1^{\mathrm{safe}}, \quad k \leftarrow 0$.
3. **while** *True* **do**

   $\quad$ # Collect new trajectories using $\pi$
4. $\quad$ $\mathcal{D} \leftarrow \emptyset$
5. $\quad$ **for** $episode \in [1, 2, ..., \delta]$ **do**
6. $\quad\quad$ $k \leftarrow k + 1$
7. $\quad\quad$ $h \leftarrow \{(s^t, a^t, \mathrm{Pr}(a^t | s^t), r^t)\}_{t=0}^T$
8. $\quad\quad$ $\mathcal{D} \leftarrow \mathcal{D} \cup (k, h)$

   $\quad$ # Split data
9. $\quad$ $\mathcal{D}_1, \mathcal{D}_2 \leftarrow \mathtt{split}(\mathcal{D})$
10. $\quad$ $\mathcal{D}_{\mathrm{train}} \leftarrow \mathcal{D}_1 \cup \mathcal{D}_{\mathrm{train}}$
11. $\quad$ $\mathcal{D}_{\mathrm{test}} \leftarrow \mathcal{D}_2 \cup D_{\mathrm{test}}$

   $\quad$ # Candidate search
12. $\quad$ $\tau \leftarrow [k + 1, ..., k + \delta]$
13. $\quad$ $\hat{\rho}^{\mathrm{lb}}(\pi), \_ \leftarrow \mathrm{PI}(\mathcal{D}_{\mathrm{train}}, \pi, \alpha/2, \tau)$
14. $\quad$ $\pi_c \leftarrow \mathrm{argmax}_\pi [\hat{\rho}^{\mathrm{lb}}(\pi) + \lambda \mathcal{H}(\pi, \mathcal{D}_{\mathrm{train}})]$

   $\quad$ # Safety test
15. $\quad$ $\hat{\rho}^{\mathrm{lb}}, \_ \leftarrow \mathrm{PI}(\mathcal{D}_{\mathrm{test}}, \pi_c, \alpha/2, \tau)$
16. $\quad$ $\_, \hat{\rho}^{\mathrm{ub}} \leftarrow \mathrm{PI}(\mathcal{D}_{\mathrm{test}}, \pi^{\mathrm{safe}}, \alpha/2, \tau)$

17. $\quad$ **if** $\hat{\rho}^{lb} > \hat{\rho}^{ub}$ **then**
18. $\quad\quad$ $\pi \leftarrow \pi_c$
19. $\quad$ **else**
20. $\quad\quad$ $\pi \leftarrow \pi^{\mathrm{safe}}$

---

**Mean future performance:** In many practical applications, it is often desirable to reduce computational costs by executing a given policy $\pi$ for multiple episodes before an update, i.e., $\delta > 1$. This raises the question regarding which episode, among the future $\delta$ episodes, should a policy $\pi$ be optimized for before execution? To address this question, in settings where $\delta > 1$, instead of choosing a single future episode's performance for optimization and safety check, we propose using the average performance across all the $\delta$ future episodes, i.e., $(1/\delta) \sum_{i=1}^{\delta} \rho(\pi, k + i)$.

**Differentiating the lower bound:** SPIN proposes a candidate policy $\pi_c$ by finding a policy $\pi$ that maximizes the lower bound $\hat{\rho}^{\text{lb}}$ of the future performance (Line 14 in Algorithm 3). To find $\pi_c$ efficiently, we propose using a differentiable optimization procedure. A visual illustration of the process is given in Figure 3.

Derivatives of most of the steps in Algorithms 1-2 can directly be taken care by modern automatic differentiable programming libraries. Hence, in the following, we restrict the focus of our discussion for describing a *straight-through* gradient estimator for sorting performed in Line 15 in Algorithm 2. Note that sorting is required to obtain the *ordered-statistics* to create an empirical distribution of $\mathtt{t}^*$ such that in Line 16 and 17 of Algorithm 2 the desired percentiles of $\mathtt{t}^*$ can be obtained.

We first introduce some notations. Let $\mathtt{t}^* \in \mathbb{R}^{B \times 1}$ be the unsorted array and $\mathtt{t}^{**} \in \mathbb{R}^{B \times 1}$ be its sorted counterpart. To avoid breaking ties when sorting, we assume that there exists $C_3 > 0$ such that all the values of $\mathtt{t}^*$ are separated by at least $C_3$. Let $\Gamma \in \{0,1\}^{B \times B}$ be a *permutation matrix* (i.e., $\forall (i,j), \ \Gamma(i,j) \in \{0,1\}$, and each row and each column of $\Gamma$ sums to 1) obtained using any sorting function such that $\mathtt{t}^{**} = \Gamma \mathtt{t}^*$. This operation has a computational graph as shown in Figure 6.

Notice that when the values to be sorted are perturbed by a very small amount, the order of the sorted array remains the same (e.g., sorting both the array $[30, 10, 20]$ and its perturbed version results in $[10, 20, 30]$). That is, if $\mathtt{t}^*$ is perturbed by an $\epsilon \to 0$, then the $\Gamma$ obtained using the sorting function will not change at all. Therefore, the derivative of $\Gamma$ with respect to $\mathtt{t}^*$ is 0 and derivative of a desired loss function $\mathcal{L}$ with respect to $\mathtt{t}^*$ is

Figure 6: Computational graph for obtaining ordered-statistics $\mathtt{t}^{**}$.

$$\frac{\partial \mathcal{L}}{\partial \mathtt{t}^*} = \Gamma^\top \frac{\partial \mathcal{L}}{\partial \mathtt{t}^{**}} = \Gamma^{-1} \frac{\partial \mathcal{L}}{\partial \mathtt{t}^{**}},$$

as for any permutation matrix, $\Gamma^\top = \Gamma^{-1}$. Therefore, derivatives are back-propagated through the sorting operation in a straight-through manner by directly performing *un*-sorting.

More advanced techniques for differentiable sorting has been proposed by Cuturi et al. [18] and Blondel et al. [8]. These methods can be leveraged to further improve our algorithm. We leave these for future work.

**Entropy regularization:** As we perform iterative safe policy improvement, the current policy $\pi$ becomes the behavior policy $\beta$ for future updates. Therefore, if the current policy $\pi$ becomes nearly deterministic then the *past performance estimates for a future policy*, which is computed using importance sampling, can suffer from high-variance. To mitigate this issue, we add a $\lambda$ regularized entropy bonus $\mathcal{H}$ in the optimization objective. This is only done during candidate policy search and hence does not impact the safety check procedure.

**Percentile CIs:** Notice that each step of the inner optimization process to search for a candidate policy $\pi_c$ requires computing multiple estimates of the pseudo standard deviation $\hat{s}^*$, one for each sample of $\mathtt{t}^*$, using wild-bootstrap to obtain the CIs. This can be computationally expensive for real-world applications that run on low-powered devices. As an alternative, we propose using the *percentile* method [11, 24] during the candidate policy search, which unlike the $\mathtt{t}$-statistic method does not require computing $\hat{s}^*$.

While percentile method can offer significant computational speed-up, the CIs obtained from it are typically less accurate than those obtained from the method that uses the $\mathtt{t}$-statistic [11, 24]. To get the best of both, (i) as searching for $\pi_c$ requires an inner optimization routine and accuracy of CIs are less important, therefore we use the percentile method, and (ii) as the safety test requires no inner optimization and accuracy of CIs are more important to ensure safety, we use the $\mathtt{t}$-statistic method.

To obtain the CIs on $\rho(\pi, k + \delta)$ using the percentile method, let $\Psi$ denote the empirical cumulative distribution function (CDF) of the pseudo performance forecasts $\hat{\rho}(\pi, k + \delta)^*$. Then a $(1 - \alpha)100\%$ CI, $[\hat{\rho}^{\text{lb}}, \hat{\rho}^{\text{ub}}]$, can be estimated as $[\Psi^{-1}(\alpha/2), \Psi^{-1}(1 - \alpha/2)]$, where $\Psi^{-1}$ denotes the inverse CDF distribution. That is, if $\rho^*$ is an array of $B$ pseudo samples of $\hat{\rho}(\pi, k + \delta)$, and $\rho^{**}$ contains its sorted ordered-statistics, then a $(1 - \alpha)100\%$ CI for $\rho(\pi, k + \delta)$ is $[\rho^{**}[(\alpha/2)B], \rho^{**}[(1 - \alpha/2)B]]$. Gradients of the lower bound from the percentile method can be computed using the same straight-through gradient estimator discussed earlier.

**Complexity analysis (space, time, and sample size):** Memory requirement for SPIN is linear in the number of past episodes as it stores all the past data to analyze performance trend of policies. As both SPIN and Baseline [57, 61] incorporate an inner optimization loop, the computational cost to search for a candidate policy $\pi_c$ before performing a safety test is similar. Additional computational cost is incurred by our method as it requires computing $(\Phi^\top\Phi)^{-1}$ and $\hat{V}$ in Algorithm 1 for time series analysis. However, note that as $\Phi^\top\Phi \in \mathbb{R}^{d \times d}$, where $d$ is the dimension of basis function and $d << k$, the cost of inverting $\Phi^\top\Phi$ is negligible. To avoid the computational cost of computing $\hat{V}$, the percentile method can be used during candidate policy search (as discussed earlier), and the t-statistic method can be used only during the safety test to avoid compromising on safety. An empirical comparison of sample efficiency of SPIN and Baseline is presented in Figure 4.

# F    Extended Empirical Details

## F.1    Domains

**Non-stationary Recommender System (RecoSys):** Online recommendation systems for tutorials, movies, advertisements and other products are ubiquitous [54, 55]. Personalizing for each user is challenging in such settings as interests of an user for different items among the products that can be recommended fluctuate over time. For an example, in the context of online shopping, interests of customers can vary based on seasonality or other unknown factors. To abstract such settings, in this domain the reward (interest of the user) associated with each item changes over time.

For $\pi^{\mathrm{safe}}$, we set the probability of choosing each item proportional to the reward associated with each item in MDP $M_1$. This resembles how recommendations would have got set by an expert system initially, such that most relevant recommendation is prioritized while some exploration for other items is also ensured.

**Non-stationary Diabetes Treatment:** This NS-MDP is modeled using an open-source implementation [68] of the U.S. Food and Drug Administration (FDA) approved Type-1 Diabetes Mellitus simulator (T1DMS) [44] for the treatment of Type-1 diabetes, where we induced non-stationarity by oscillating the body parameters (e.g., rate of glucose absorption, insulin sensitivity, etc.) between two known configurations available in the simulator. Each step of an episode corresponds to a minute in an *in-silico* patient's body and is governed by a continuous time non-linear ordinary differential equation (ODE) [44].

Notice that as the parameters that are being oscillated are inputs to a non-linear ODE system, the exact trend of performance for any policy in this NS-MDP is unknown. This more closely reflects a real-world setting where Assumption 1 might not hold, as every policy's performance trend in real-world problems cannot be expected to follow *any* specific trend *exactly*–one can only hope to obtain a coarse approximation of the trend.

To control the insulin injection, which is required for regulating the blood glucose level, we use a parameterized policy based on the amount of insulin that a person with diabetes is instructed to inject prior to eating a meal [5]:

$$\text{injection} = \frac{\text{current blood glucose} - \text{target blood glucose}}{CF} + \frac{\text{meal size}}{CR},$$

where 'current blood glucose' is the estimate of the person's current blood glucose level, 'target blood glucose' is the desired blood glucose, 'meal size' is the estimate of the size of the meal the patient is about to eat, and $CR$ and $CF$ are two real-valued parameters that must be tuned based on the body parameters to make the treatment effective. We set $\pi^{\mathrm{safe}}$ to a value near the optimal $CR$ and $CF$ values for MDP $M_1$. This resembles how the values would have got set during a patient's initial visit to a medical practitioner.

## F.2    Baseline

For a fair comparison, Baseline used for our experiments corresponds to the algorithm presented by Thomas et al. [57], which is also a type of Seldonian algorithm [61]. While this algorithm is also designed to ensure safe policy improvement, it assumes that the MDP is stationary. Specifically, during the safety test it ensures that a candidate policy's performance is higher than that of $\pi^{\mathrm{safe}}$'s by computing CIs on the *average* performance over the past episodes.

## F.3 Hyper-parameters

In Table 3, we provide hyper-parameter (HP) ranges that were used for SPIN and Baseline for both the domains. As obtaining optimal HPs is often not feasible in practical scenarios, algorithms that ensure safety should be robust to how an end-user sets the HPs. Therefore, we set the hyper-parameters within reasonable ranges and report the results in Figure 4. These results are aggregated over the *entire* distribution of hyper-parameters, and *not* just for the best hyper-parameter setting. This choice is motivated by the fact that best performances can often be misleading as it only shows what an algorithm *can* achieve and not what it is *likely* to achieve [32].

For both RecoSys and Diabetes, we ran 1000 HPs per algorithm, per speed, per domain. For RecoSys, we ran 10 trials per HP and 1 trial per HP for diabetes treatment as it involves solving a continuous time ODE and hence is relatively computationally expensive. For experiments, the authors had shared access to a computing cluster, consisting of 50 compute nodes with 28 cores each.

| Algorithm | Hyper-parameter | Range |
|---|---|---|
| SPIN & Baseline | $\alpha$ | 0.05 |
| SPIN & Baseline | $\delta$ | $\{2, 4, 6, 8\}$ |
| SPIN & Baseline | $N$ | $\delta \times$ `uniform`$(\{2, 5\})$ |
| SPIN & Baseline | $\eta$ | $10^{-1}$ |
| SPIN & Baseline | $\lambda$ (RecoSys) | `loguniform`$(5 \times 10^{-5}, 10^0)$ |
| SPIN & Baseline | $\lambda$ (Diabetes) | `loguniform`$(10^{-2}, 10^0)$ |
| SPIN & Baseline | $B$ (candidate policy search) | 200 |
| SPIN & Baseline | $B$ (safety test) | 500 |
| SPIN | $d$ | `uniform`$(\{2, 3, 4, 5\})$ |

Table 3: Here, $N$ and $\eta$ represents the number of gradient steps, and the learning rate used while performing Line 14 of Algorithm 3. The dimension of Fourier basis is given by $d$. Notice that $d$ is set to different values to provide results for different settings where SPIN is *incapable* of modeling the performance trend of policies exactly, and thus Assumption 1 is violated. This resembles practical settings, where it is not possible to exactly know the true underlying trend–it can only be coarsely approximated.

## F.4 Plot Details

In Figure 4, the plot on the bottom-left corresponds to how often unsafe policies were executed during the process whose learning curves were plotted in Figure 4 (top-left). It can be seen that SPIN remains safe almost always. The middle and the right plots in the top row of Figure 4 show the normalized performance improvement over the known safe policy $\pi^{\text{safe}}$. The middle and the right plots in the bottom row of Figure 4 show how often unsafe policies were executed.

Note that the performance for any policy $\pi$ is defined in terms of the expected return. However, for the diabetes domain, we do not know the exact performances of any policy–we can only observe the returns obtained. Therefore, even when an `alg` selects $\pi^{\text{safe}}$, it is not possible to get an accurate estimate of its safety violation rate by directly averaging returns observed using a finite number of trials. To make the evaluation process more accurate, we use the following evaluation procedure.

Let a policy $\pi$ be 'unsafe' when $\rho(\pi, k + \delta) < \rho(\pi^{\text{safe}}, k + \delta)$, and let $\pi_c$ denote policies not equal to $\pi^{\text{safe}}$, then,

$$\Pr(\texttt{alg}(\mathcal{D}) = \text{unsafe}) = \Pr(\pi_c = \text{unsafe}|\texttt{alg}(\mathcal{D}) = \pi_c) \Pr(\texttt{alg}(\mathcal{D}) = \pi_c)$$
$$+ \Pr(\pi^{\text{safe}} = \text{unsafe}|\texttt{alg}(\mathcal{D}) = \pi^{\text{safe}}) \Pr(\texttt{alg}(\mathcal{D}) = \pi^{\text{safe}})$$
$$\overset{\textbf{(a)}}{=} \Pr(\pi_c = \text{unsafe}|\texttt{alg}(\mathcal{D}) = \pi_c) \Pr(\texttt{alg}(\mathcal{D}) = \pi_c),$$

where **(a)** holds because $\Pr(\pi^{\text{safe}} = \text{unsafe}) = 0$. Therefore, to evaluate whether $\texttt{alg}(\mathcal{D})$ is unsafe, for each episode we compare the sample average of returns obtained whenever $\texttt{alg}(\mathcal{D}) \neq \pi^{\text{safe}}$ to the sample average of returns observed using $\pi^{\text{safe}}$, multiplied by the probability of how often $\texttt{alg}(\mathcal{D}) \neq \pi^{\text{safe}}$.

## Footnotes

[2]Note that the original result by Achiam et al. [2] bounds the change in distribution between two different policies under the same dynamics. Here, we have modified the property for our case, where the policy is fixed but the dynamics are different.

[3]When $(\alpha/2)B$ or $(1-\alpha/2)B$ is not an integer, then `floor` or `ceil` operation should be used, respectively.