[Reviews · NeurIPS 2020]

Review 1

Summary and Contributions: In this paper, the authors introduce a novel model-free, policy improvement-based algorithm, for smooth non-stationary Markov decision processes (NS-MDP), focusing on safety guarantees of their method. The method relies heavily on Assumption 1 (Smooth performance), implicitly assumed in [51], which enables the treatment of the off-policy evaluation (OPE) in the NS-MDP as a time-series forecasting (TSF) problem. The authors introduce safe policy improvement for NS-MDPs, which they term SPIN, which, under Assumption 1, iterates between a policy evaluation step and a policy improvement step. Importance sampling is used for OPE according to past evaluation samples. Then TSF is applied to estimate future performance, while wild bootstrapping is used to obtain uncertainty estimates for future performance. A learnable safe policy is used when the performance estimates are below the safe policy, otherwise, a policy improvement step is made. The experiments on a non-stationary recommender system, RecoSys, and a non-stationary diabetes treatment simulated environment suggest that the method outperforms methods that ignore non-stationarities but strive for safety only.

Strengths: This paper address a very important problem in reinforcement learning (RL), the usually false assumption of stationarity. As highlighted by the authors, most real-world settings are intrinsically non-stationary and hence ignoring this can lead to catastrophic outcomes. The authors provide a way time-series-forecasting (TSF) can be used to enable RL in non-stationary settings since TSF has been used for non-stationaries extensively. This link can encourage future research in this direction. To the best of my knowledge, the method introduced in this paper is novel and the empirical results suggest that it is successful in (smooth) non-stationary settings.

Weaknesses: The method relies heavily on Assumption 1 (Smooth performance), which I do not have good intuition to what that translates, in terms of drift in transition dynamics and reward function. I wonder if there is a direct connection to a Lipschitz smoothness assumption. Although the empirical results and the theoretical justifications of the introduced method, SPIN, are sound, the lack of an ablation study makes it hard to credit assign the contribution of each sub-component to the final performance improvement. For example, how important is practice (a) the data-splitting [lines 235-237, 243-244]; (b) searching for the highest lower confidence bound [lines 219-220] over, e.g. the expected performance; (3) the wild bootstrap. It would be great to see an ablation study where starting from the Baseline (or a simplified backbone) the authors build methods towards full SPIN and observe which decisions result in what gains, and also which combinations were the most impactful. Also how about methods that only consider non-stationarities, without striving for safety?

Correctness: In my opinion, the claims and method are correct. They are either supported by empirical evidence, or references or analytical derivations. The experimental protocol is also well described and doesn’t have any obvious flaws.

Clarity: The paper is well written. It’s self-contained and comprehensive.

Relation to Prior Work: The authors discuss how their work differs from previous contributions as well as provide the relevant background (both in Section 2 and in the Appendix) so that their method is self-contained and understood. However, I wonder how this work relates to (a) lifelong reinforcement learning and (b) zero-shot adaptation literature, which either explicitly or implicitly make non-stationarity assumptions and devise algorithms that address similar challenges with the ones the authors face. Also, some of the high-level decisions made by the authors reminded me of some prior work in “safe imitation learning” [A]. Also, it’s not very clear if the authors credit Assumption 1 to [51] or if they claim to have introduced it in this work. [A] Zhang et al. (2016) Query-Efficient Imitation Learning for End-to-End Autonomous Driving.

Reproducibility: Yes

Additional Feedback:


Review 2

Summary and Contributions: This paper studies safety in non-stationary MDP, defining safety as as not reducing performance with respect to a current "safe policy." Prior approaches have assumed stationary MDPs. Motivations come from settings such as diabetes, sepsis, and e-commerce. The model considers an exogenous sequence over MDPs and makes use of time-series forecasting style approaches to estimate the performance of distinct policies. Crucially, safety is only possible with a smooth transition between environments, so that inference can be done about relative policy quality (either P or R moving slowly, or the payoff of a policy changing slowly enough). The technical contribution is to estimate the performance of a counterfactual policy via importance sampling, following Precup and Thomas, and "wild bootstrap"to get confidence intervals. The "Seldonian" framework from Thomas et al. (2019) is used to perform sequential hypothesis testing, wuth a sophisticated test, search and data splitting approach.

Strengths: The paper is clear, comprehensive, and convincing. Beyond new technical results (Theorem 1, Theorem 2), the SPIN approach is carefully evaluated in simulation in the settings of recommeder systems and diabetes treatment.

Weaknesses: None noted. After the rebuttal: thanks also to the authors for their careful responses to the comments from other reviewers.

Correctness: yes, appears correct.

Clarity: yes, very clear.

Relation to Prior Work: yes, very good.

Reproducibility: Yes

Additional Feedback:


Review 3

Summary and Contributions: Update: Thanks the authors for the detailed feedback. All my concerns are addressed, and I am happy to maintain my judgement of accepting this paper. The paper consider RL with safety constraint (by comparing to a known safe policy). The authors propose algorithms for this setting, and perform empirical analysis.

Strengths: This is by far the first paper that considers safety constraints in RL. The authors provide detailed steps to build Rl algorithms that performs no worse than a baseline policy with high probability

Weaknesses: 1. It appears that the non-stationarity considered in this paper is governed by a underlying linear model, and seems like the phi(i)'s are known ahead of time (please correct me if I am wrong). 2. Also, seems like [8] is a very closely related setting, and the solutions are pretty related. It would be helpful if the authors can explain the key differences in techniques.

Correctness: It looks correct to me.

Clarity: It is well written.

Relation to Prior Work: Some related works on conservative bandit exploration is not mentioned. For example: Abbas Kazerouni, Mohammad Ghavamzadeh, Yasin Abbasi-Yadkori, Benjamin Van Roy (2016). "Conservative Contextual Linear Bandits."

Reproducibility: Yes

Additional Feedback:


Review 4

Summary and Contributions: This paper proposes a seldonian framework for safe policy adaptation in nonstationary MDPs. I find the paper interesting and the approach also interesting. However, I am a bit unsure about the overall architecture and the empirical evaluation.

Strengths: The paper proposes algorithms for nonstationary MDPs, which are a challenging and open area in RL. The findings would showcase how algorithms could be designed in the presence of nonstationarity using the seldonian framework, which in itself is a recent framework. The algorithm seems to be based on the Seldonian framework, which uses the pseudo samples. The pseudo samples are generated from regressed models of the existing samples. The authors do note that this can only work when the MDP satisfies some smoothness properties.

Weaknesses: It is not clear how many real world problems would satisfy these properties, and for those cases, would adding a state to the MDP address the issue of nonstationarity?

Correctness: The theory and experiments seem correct

Clarity: yes, the paper is well written. The assumptions are clearly stated.

Relation to Prior Work: Yes

Reproducibility: Yes

Additional Feedback: I found the paper hard to read and the idea of creating pseudo samples is not very crystal still to me. An algorithm would have helped here, since this seems like the key thing that would affect the algorithm. What happens if the pseudo samples fail? How much data is necessary before this can be done safely? I have gone through other reviews and feedback and stay with my recommendation.

[Author Response · NeurIPS 2020]

We thank all reviewers for their insightful suggestions. In the following, we address all the questions in order.

**R1:** " direct connection" - This is an interesting question! We do not think that Assumption 1 will have a direct relation in terms of Lipschitz smoothness (which is an extremely local property). For example, to infer the performance of $\pi$ on $M_k$, no *additional* information is gained by knowing performances of $\pi$ on MDPs $\{M_i\}_{i=1}^{k-2}$ when the Lipschitz constraint and the performance of a policy on MDP $M_{k-1}$ are known. This is unlike our assumption, where data from the past MDPs $\{M_i\}_{i=1}^{k-1}$ can be informative towards inferring the performance of a policy on $M_k$.

"data/split", "highest lower" - We will clarify that our consideration of the lower bound for optimization was based on similar techniques used in the literature [12, 21, 52], and is not a primary contribution of our work. In Table 1, we provide an ablation study for RecoSys, for all the speeds $(0, 1, 2, 3)$. All other experimental details are the same as in Appendix E.3, except for (iv), where mean performance is optimized for instead of the lower bound. It can be seen that the safety violation rate of SPIN is robust against such hyper-parameter changes. Although, it is worth noting that too small a test-set can make it harder to pass the safety-test for executing a $\pi_c \neq \pi^{\text{safe}}$, hence performance improvement is marginally low in (i). Thank you for suggesting these experiments to improve the paper, we will include these results in the appendix.

"without striving for safety?" - If the safety check procedure for a policy's performance on a non-stationary MDP (which is one of the primary contributions of our work) is removed, then the results can be catastrophic, as can be seen in (v).

"wild bootstrap" - Time series literature is vast and it is not obvious to us which other method would be more suitable to address the challenges mentioned in Lines 147–156. A detailed discussion is provided in Appendix C.2 and C.3 regarding why several popular techniques would be ill-suited.

"lifelong", "zero-shot", "safe imitation" - Thank you for pointing these out. We will discuss these in the main paper.

"credit Assumption 1" - While we did formalize the implicit assumption made by [51] in the context of reinforcement learning, this type of assumption is popular in time series literature [6]. We will discuss this in the paper.

|       |                              | train-test | 0   | 1   | 2   | 3   | 0   | 1    | 2    | 3    |
|-------|------------------------------|------------|-----|-----|-----|-----|-----|------|------|------|
| (i)   | SPIN                         | 75%-25%    | .56 | .22 | .17 | .14 | 0.0 | 3.6  | 5.1  | 5.4  |
| (ii)  | SPIN                         | 25%-75%    | .48 | .29 | .21 | .19 | 0.0 | 4.6  | 6.5  | 7.0  |
| (iii) | SPIN (Fig. 4)                | 50%-50%    | .62 | .28 | .21 | .18 | 0.0 | 4.7  | 6.4  | 6.6  |
| (iv)  | SPIN-mean                    | 50%-50%    | .70 | .28 | .24 | .19 | 0.2 | 4.9  | 6.3  | 7.1  |
| (v)   | Non-stationary + No safety   | 100%-0%    | .73 | .22 | .16 | .19 | 9.4 | 37.6 | 40.2 | 38.6 |
| (vi)  | Stationary + Safety (Fig. 4) | 50%-50%    | .85 | .12 | .07 | .07 | 0.0 | 19.8 | 15.3 | 11.9 |

Table 1: (Left) Algorithm. (Middle) Improvement over $\pi^{\text{safe}}$. (Right) Safety violation percentage.

**R2:** Thank you for your support!

**R3:** "underlying linear model"- We will clarify this point of confusion in the paper. Yes, Assumption 1 requires the trend (policy's performance over time) to be a linear function of the features, $\phi$, which are known ahead of time. We will state this explicitly in the paper, while reminding readers that this allows for non-linear functions when $\phi$ are non-linear. Additionally, we will discuss the flexibility offered by the Fourier basis for modeling a wide-class of trends [6], and emphasize Lines 271–274 to indicate that our experimental section also includes a domain (Diabetes treatment) where Assumption 1 is violated.

"explain the key differences" - We will clarify lines 57–60 to highlight that our paper extends prior work [8,51] to quantify uncertainty about a policy's future performance and to provide safety guarantees.

"conservative bandit exploration" - Thank you for pointing this out. We will include this in the main paper.

**R4:** "how many real world problems would satisfy these properties" - This is a good point: We should have, and will, discuss around Lines 345–347 how a practitioner can or should apply our method. Like any time-series forecasting problem, before applying our method goodness-of-fit tests [10] can be used by practitioners to check whether Assumption 1 is reasonable. For example, notice that Fig. 5 in [51] shows that this assumption is reasonable for a real digital marketing dataset. Furthermore, we will discuss how this is at least a step in the right direction: standard methods that make stationarity assumptions correspond to our method with $\phi(s) = [1]$ always (fitting a horizontal line). Even if Assumption 1 is not satisfied exactly, if the trend has an overall pattern, it is likely better to account for this overall pattern than to resort back to standard methods (fitting a horizontal line).

"algorithm would have helped" - Due to space constraints, the algorithm was deferred to Appendix D.

"pseudo samples fail", "How much data is necessary"- These are great questions! Unfortunately, there is no exact answer. Bootstrap methods provide approximate bounds and their failure rate is typically of the order $O(n^{-p/2})$, where $p \in [1, 3]$ and $n$ is the number of samples. Lines 662–668 in the appendix provide a more detailed discussion.

[Meta-Review · NeurIPS 2020]

With reviewer scores of (9, 7, 6, 6) this submission is overwhelmingly likely to be accepted. The submission describes a novel method combining Safe Policy Improvement in Non-stationary (SPIN) MDPs. The method alternates between Policy Evaluation and Policy Improvement with safety guarantees. The reviewers generally agree that the writing is clear, presents new technical results (based on Assumption #1) and good empirical evaluation in two domains: recommender systems.